# EXPERIMENTAL DESIGN FOR NONSTATIONARY OPTIMIZATION

## ABSTRACT

Traditional methods for optimizing neural networks often struggle when used to train networks in settings where the data distributions change, and plasticity preservation methods have been shown to improve performance in such settings (e.g. continual learning and reinforcement learning). With the growing interest in nonstationary optimization and plasticity research, there is also a growing need to properly define experimental design and hyperparameter search protocols to enable principled research. Each new proposed work typically adds several new hyperparameters and makes many more design decisions such as hyperparameter selection protocols, evaluation protocols, and types of tasks examined. While innovation in experiment design is important, it is also necessary to (1) question whether those innovations are leading to the best progress and (2) have standardized practices that make it easier to directly compare to prior works. In this paper, we first perform an extensive empirical study of over 27,000 trials looking at the performance of different methods and hyperparameters across different settings and architectures used in the literature to provide an evaluation of these methods and the hyperparameters they use under similar experimental conditions. We then examine several core experiment design choices made by the community, affirming some while providing evidence against others, and provide concrete recommendations and analysis that can be used to guide future research.

## 1 INTRODUCTION

Deep learning has seen success across a wide range of tasks and datasets. The trend of larger models and larger amounts of data, however, suggests the need to be able to update our models in an online fashion on changing tasks and datasets. Most of our currently successful machine learning systems were developed with training methods that assumed a static training distribution. When the same training methods are applied in settings with changing, or *nonstationary*, data distributions, they struggle to achieve the same level of performance (Ash & Adams, 2020; Abbas et al., 2023-08-22/2023-08-25; Nikishin et al., 2022). In fact, in settings where nonstationarity is inherent either by construction (e.g. continual learning) or because the solution methods require it (e.g. reinforcement learning), many solutions instead focus on trying to transform the optimization problem into something as stationary as possible (Mnih et al., 2015; Rolnick et al., 2019; Chaudhry et al., 2019b).

Recently, there has been a growing recognition of the limitations of traditional optimization methods when applied to nonstationary problems. Specifically, recent works have looked at the phenomena where neural networks get worse at both reducing training error and improving generalization performance when exposed to different distributions than what they were trained on initially, problems often collectively referred to as *loss of plasticity*. There are many different methods proposed to mitigate this loss of plasticity, and mostly can be described as one of architectural (Abbas et al., 2023-08-22/2023-08-25; Lyle et al., 2023-07-23/2023-07-29), regularization (Lyle et al., 2023-07-23/2023-07-29; Lewandowski et al., 2024), or resetting (Lyle et al., 2024b; Dohare et al., 2021) based.

The works introducing these methods also introduce many other experimental design choices. As a consequence of the decentralized nature of research, these choices include everything from the types of nonstationarities used to evaluate the methods to the protocols used to select hyperparameters, and often, these choices are hidden or unclear. In settings that require

nonstationary optimization such as continual learning (Cha & Cho, 2024) and reinforcement learning (Henderson et al., 2018; Obando-Ceron et al., 2024), these choices have been shown to have a huge impact on the evaluation of different methods: simply tuning an old method on a new setting can result in new state-of-the-art results (Chaudhry et al., 2019b; van Hasselt et al., 2019; Schwarzer et al., 2023).

With that in mind, our work examines a representative sample of plasticity preserving methods across several different types of nonstationary settings and architectures that have been used in the literature. Specifically, we look at permuted input, label shuffled, and noisy label versions of the CIFAR-10 and CIFAR-100 (Krizhevsky, 2009) datasets, using Multilayer Perceptrons (MLPs) and ResNets (He et al., 2016). We choose settings and architectures that the community already works with so that our findings can be directly useful to people working on these problems. We first perform an extensive hyperparameter sweep for each of these methods and evaluate them with a consistent setup to evaluate which methods work well on different architectures and settings. We assess the transferability of these methods and the impact of their hyperparameters on their performance.

We then further explore the experimental design decisions that need to be made when creating an experiment for nonstationary optimization such as the number of seeds to use when evaluating a hyperparameter configuration, or which metric to use to evaluate a model. Our goal is to not only provide the research community with empirical results to guide the decision making that is often done based on arbitrary intuition, but also enable research for groups with access to fewer resources by telling them where to focus their resources.

To summarize, our contributions include a comprehensive evaluation of widely used current plasticity preserving methods and their hyperparameters across several types of nonstationarities and architectures, and an empirical evaluation of various experimental design decisions that go into doing research on nonstationary optimization including:

- Which hyperparameter selection protocol to use?
- Whether we should be optimizing for both training and test accuracy when doing nonstationary optimization research.
- How can we do a resource efficient hyperparameter search?

As part of our work, upon acceptance we will also be releasing the code and all of the intermediate results for all the evaluations described in the rest of this paper as a base for future research.

## 2 RELATED WORK

### 2.1 PLASTICITY AND NONSTATIONARY OPTIMIZATION

After training on a task, neural networks have been shown to struggle to adapt to new data distributions in both the continual supervised learning (Dohare et al., 2024) and the reinforcement learning settings (Nikishin et al., 2022; Abbas et al., 2023-08-22/2023-08-25). In fact, Ash & Adams (2020) show that even networks pretrained on a subset of the task data underperform relative to randomly initialized (untrained) networks when they are later trained on the full dataset. This type of nonstationarity has been shown to affect the network's ability to reduce both the training error (Dohare et al., 2024) and the generalization, or testing error (Ash & Adams, 2020; Lee et al., 2024b). We will refer to these as trainability and generalizability in the rest of the paper. The term *loss of plasticity* usually refers to the first problem and sometimes the second problem. We will use "loss of plasticity" as the umbrella term for both problems and use "trainability" and "generalizability" when referring to the specific problems. Several different works have started exploring mechanisms and causes for plasticity loss (Lyle et al., 2023-07-23/2023-07-29; Lewandowski et al., 2024; Lyle et al., 2024b; Kumar et al., 2023). There has also been a growing literature showing that addressing plasticity issues can lead to significant performance improvements not only in the standard continual learning settings, but also for reinforcement learning agents (Abbas et al., 2023-08-22/2023-08-25; Schwarzer et al., 2023; Lee et al., 2024a).

Plasticity loss mitigation measures usually fall into one of three categories: regularization, architectural, or resetting. *Regularization* based approaches involve adding a penalty or constraining the parameters in some way, such as regularizing towards the initialization (Kumar et al., 2023;

(a) The three different distribution shifts described in Section 3.1.

(b) The three hyperparameter selection protocols described in Section 3.3.

Figure 1: The task shifts and hyperparameter selection protocols used in our study.

Lewandowski et al., 2024) or towards smaller weight norm (Lyle et al., 2023-07-23/2023-07-29). *Architectural* approaches involve modifying the neural network architecture itself, such as adding normalization layers (Lyle et al., 2024b;a) or different activations (Abbas et al., 2023-08-22/2023-08-25; Lee et al., 2024a). Finally, *Resetting* based approaches involve reinitializing or perturbing the network weights in ways to reintroduce plasticity (Ash & Adams, 2020; Lee et al., 2024b; Sokar et al., 2023; Abbas et al., 2023-08-22/2023-08-25).

## 2.2 HYPERPARAMETER SEARCH IN NONSTATIONARY SETTINGS

Because hyperparameters are intimately tied to how we use and evaluate any machine learning algorithm, doing proper and equitable hyperparameter optimization is critical to properly compare against prior work and make progress as a research community. For example, in reinforcement learning (RL), simply tuning an existing baseline created a new state-of-the-art result on the Atari100k benchmark (Kaiser et al., 2019). Given the complexity and resource intensiveness of RL, further work explored how to best and most efficiently tune hyperparameters for RL algorithms, including examining reproducibility issues (Henderson et al., 2018), how to properly use multiple seeds to evaluate an algorithm (Agarwal et al., 2021), and the consistency of selected configurations in a search (Obando-Ceron et al., 2024).

Hyperparameter search in continual supervised learning suffers from similar issues of complexity and resource requirements, with the added ambiguity of the criterion that should actually be used for the selection and evaluation of a hyperparameter configuration. Previous works on hyperparameter optimization in continual supervised learning focused on settings trying to mitigate forgetting, i.e. performance degradation on tasks seen earlier in training, while learning new tasks. A common approach is to set aside a portion of the training data from each incoming task as a validation set, select the hyperparameters based on some aggregate metric on the validation sets across all tasks, and then use the test set metrics as the final evaluation (Masana et al., 2023). Chaudhry et al. (2019a); Cha & Cho (2024) propose tuning hyperparameters on one sequence of tasks and evaluating on a different sequence of tasks. Lee et al. (2024c) explore different protocols where hyperparameters were chosen after just a single task or were dynamically adapted after each task.

Our work evaluates several of these proposed protocols in the context of nonstationary optimization, where we only try to maximize the model's performance on the current task and do not attempt to maintain performance on previous tasks. Since several previous works in this area are unclear on some or all of the details of the experimental setup such as which hyperparameter selection protocol was used or how many seeds were used, we also reimplement a representative sample of previous works and evaluate them on a consistent setup.

## 3 BENCHMARKING SETUP

In this section, we outline the details of the different hyperparameter searches we conducted. We outline the methods we analyzed in Section 3.2. For each method, we randomly sample 40 configurations from the full search space for that method, as grid search would become prohibitively expensive for some methods, and other methods like Bayesian search would impose a temporal order on the sampling of the configurations that would make the analysis in Section 4 more difficult.

We study two different architectures, a 3 layer MLP with 128 hidden units at each layer and a ResNet-18 (He et al., 2016). These architectures not only allow us to examine the methods at different scales with different layer types, but they are also commonly used in the literature, making our analysis more valuable for the community. We do a separate search for each combination of method, architecture, and task stream. For each configuration sampled, we evaluate $n = 20$ seeds for MLP runs and $n = 10$ seeds for ResNet-18 runs, with each seed having a different task stream as well as model initialization.

The community has targeted both training accuracy (Dohare et al., 2021; Kumar et al., 2023; Lewandowski et al., 2024; Lyle et al., 2024b) and test accuracy (Lee et al., 2024b; Elsayed et al., 2024) as measures of interest when doing plasticity research. We study both, and particularly the relation between them in Sections 4 and 4.3, and focus on test accuracy in the rest of the study. For the Shuffled and Permuted settings, we compute the average accuracy across all tasks, while for the Noisy setting, we only use the testing accuracy on the final task.

### 3.1 Nonstationarities

Each run in our setting involves training on a series of $m$ tasks in sequence, with task $k$ involving learning on dataset $\mathcal{D}_k = \{(x_i^k, y_i^k)_{i=1}^{n_k}\}$. The goal of the learner is to maximize performance on task $k$, without trying to preserve performance on tasks $1 \ldots k-1$. Several different nonstationarity types have been proposed and studied in the plasticity literature. Most of them involve taking some base dataset and applying some transformation. We will take a similar approach, with CIFAR-10 and CIFAR-100 (Krizhevsky, 2009) being used as the base datasets for the MLP and the ResNet-18 experiments respectively. In our study, we focus on the following three transformations (also shown in Figure 1a):

**Shuffled Label** Our first transformation is the shuffled label transformation, where each task is created by remapping the labels of the original dataset to new labels. Specifically, given the original dataset $\mathcal{D} = \{(x_i, y_i)_{i=1}^{n}\}$ and a randomly generated permutation function $\mathcal{P}_y^k : \mathcal{Y} \to \mathcal{Y}$ that remaps the label space, task $k$ involves learning on $\mathcal{D}_k = \{(x_i, \mathcal{P}_y^k(y_i))_{i=1}^{n}\}$. We prefer this output transformation to the other commonly used output transformation where each sample is assigned a random label as this allows us to probe the network's ability to maintain generalizability and not just trainability. We set the number of tasks $m = 100$ and $m = 30$ for MLP and ResNet-18 experiments respectively.

**Permuted Input** Our second transformation is similar to the first, except instead of permuting the output space, we permute the input space. This is a commonly used benchmark in continual learning (Goodfellow et al., 2013), where given $\mathcal{D}$ and a randomly generated permutation function $\mathcal{P}_x^k : \mathcal{X} \to \mathcal{X}$ that permutes the locations of the pixels in input space, task $k$ involves learning on $\mathcal{D}_k = \{(\mathcal{P}_x^k(x_i), y_i)_{i=1}^{n}\}$. Similar to the shuffled label setup, we set the number of tasks $m = 100$ and $m = 30$ for MLP and ResNet-18 experiments respectively.

**Noisy to Clean Label** Our final transformation was proposed in (Lee et al., 2024b). Assuming there are $m$ total tasks in the task sequence, the dataset is split into $m$ equal chunks, and then label noise is applied to each chunk, going from a high level of noise at the beginning of training to a clean chunk at the end of training. Given $\mathcal{D}$, a corruption function $\mathcal{T} : \mathcal{Y} \times P \to \mathcal{Y}$, and corruption probability $p^k$ for task $k$, task $k$ involves learning on $\mathcal{D}_k = \{(x_i, \mathcal{T}(y_i, p^k))_{i=(k-1) \cdot \lfloor \frac{n}{m} \rfloor}^{k \cdot \lfloor \frac{n}{m} \rfloor}\}$. For both MLP and ResNet-18 experiments, we split the datasets into 10 chunks and linearly interpolate the corruption probability from .5 to 0 over the course of the task sequence.

### 3.2 Methods

***Online*** This method does no plasticity preserving intervention other than the default L2 regularization.

***L2 Init*** (Kumar et al., 2023) This method, also known as regenerative regularization, replaces the default L2 regularization (towards $\vec{0}$) with an L2 regularization towards the network initialization.

***LayerNorm*** This method simply adds Layer Normalization (Ba et al., 2016) before each ReLU (Agarap, 2019) activation.

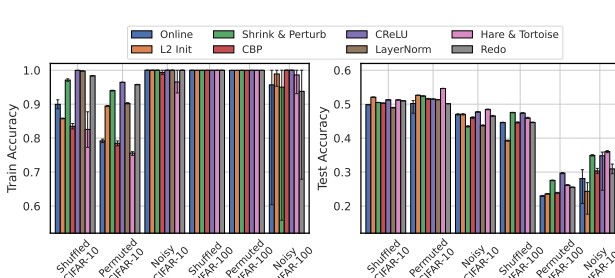 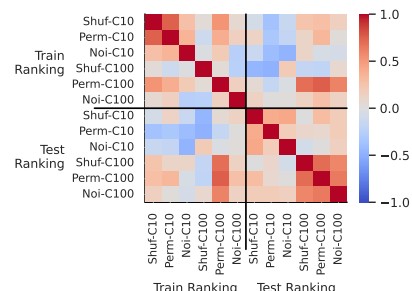

(a) Training and testing performance of different plasticity preserving methods from the literature across different distribution shifts and architectures. Error bars represent 95% confidence intervals.

(b) Kendall rank correlation coefficient of the method rankings generated from the performances on different distribution shifts. 1.0 is perfectly correlated and -1.0 is perfectly anti-correlated.

Figure 2: We present the performance of methods from the literature on settings representing different architectures, distribution shifts, and datasets (left), as well as how well the the method rankings for each setting correlate with each other (right).

**CReLU** (Abbas et al., 2023-08-22/2023-08-25) This method converts each ReLU activation function into a (Concatenated ReLU) CReLU activation function. CReLU concatenates $[\mathrm{relu}(x), \mathrm{relu}(-x)]$, which increases the size of the network compared to other methods. We do not control for the number of parameters in our study.

**Redo** (Sokar et al., 2023) This method periodically finds the neurons with low activation absolute values, and resets their incoming weights randomly and their outgoing weights to 0.

**Hare & Tortoise** (Lee et al., 2024b) This method maintains two copies of the network, a *Hare* network that is trained on the data and a *Tortoise* network that is an exponentially moving average of the *Hare* network. The parameters of the *Hare* network are periodically reset back to the parameters of the *Tortoise* network.

**Shrink & Perturb** (Ash & Adams, 2020) This method multiplies the network parameters by a shrinkage factor $p < 1$, and then perturbs the parameters with scaled noise sampled from the same distribution as the network initialization.

**CBP** (Dohare et al., 2024) This method computes a utility function for each neuron and resets the weights connected to low utility neurons (in a similar fashion to *Redo*) if the neuron had not been reset in a while.

For each method, we search jointly over the method hyperparameters, as well as the optimizer (SGD vs Adam), the learning rate, the L2 regularization penalty, and if the optimizer chosen was Adam, the values for $\beta_1, \beta_2$, and $\epsilon$. See Appendix A for the full search spaces.

### 3.3 Selection Protocols

In standard machine learning, selecting a hyperparameter configuration is fairly straightforward. To avoid overfitting to the test set, practitioners perform k-fold cross-validation where they split the dataset into $k$ pieces, retraining $k$ times each time leaving out one of the pieces to use for evaluation (Hastie et al., 2009). With standard deep learning, this becomes more difficult as retraining is expensive, and so we simply set a piece of the training data aside as validation to use for selecting the best configuration, then evaluating on the test set. With continual learning, this procedure becomes ambiguous. Since we now have $m$ different train, test, and validation datasets, as well as different task sequences between $n$ seeds, how do you select and evaluate your configurations?

We describe three protocols proposed in the literature for continual learning (Figure 1b). Each protocol consists of a hyperparameter configuration *selection* procedure used to select amongst the different configurations being evaluated in the search and a final *evaluation* procedure, which is used to report the performance of the method and compare it to other methods.

**Protocol 1:** A commonly used protocol in continual learning (Masana et al., 2023) is to split the training dataset for each incoming task into a training dataset and validation dataset. The average validation metric (loss, accuracy, or any other metric of interest) across the different seeds and tasks is used to select the best configuration, and the average test metric is reported for evaluation. The selection and evaluation are done on the same task sequences.

**Protocol 2:** An alternative is to use multiple streams of tasks in the protocol (Chaudhry et al., 2019a; Cha & Cho, 2024). Specifically, the test metric on one task sequence is used to select the configuration, and the test metric on another task sequence is reported as the final evaluation. Cha & Cho (2024) claim that Protocol 1 can overfit to the specific task sequence used, and this procedure can mitigate that risk. In our study, each seed results in different transformations being applied and thus represents a different task sequence.

**Protocol 3:** Finally, (Mesbahi et al., 2024) propose a protocol in the context of continual reinforcement learning where the test metric on the first $k\%$ of training is used for selection, and the test metric on the rest of training is used for evaluation, claiming that this is a more realistic and challenging protocol for continual learning.

## 4 A STUDY IN PLASTICITY

In this section, we analyze the results of the study described in section 3. We answer questions about widely used methods for plasticity loss mitigation from the literature, as well as questions on the practices used to evaluate them.

### 4.1 COMPARING PROTOCOLS

We first compare the effectiveness of the protocols described in Section 3.3. A further discussion of Protocol 3 will be done in Section 4.5. We first divide the seeds/task sequences available for each configuration. We use half to create a "held out" ranking of the methods by taking the configuration of each method with the highest average test accuracy across those seeds. With the other half of the seeds, we do model selection and evaluation based on each of the protocols. We compute our estimates using statistical bootstrapping with 1000 trials by resampling the seeds used to do model selection/evaluation. Based on the evaluations from each protocol, we rank the performance of each method and compare those rankings to the held out ranking. For Protocol 3, we use the first 20% of tasks in the sequence for selection and the latter 80% for evaluation.

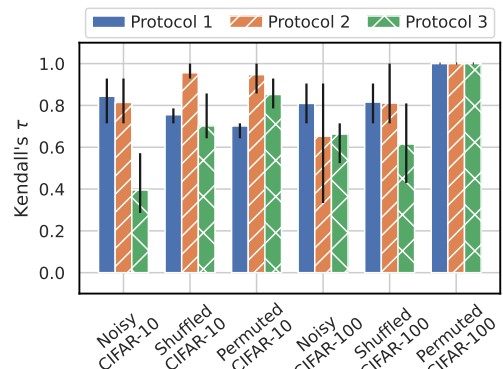

Figure 3: A look at how the method rankings generated by the protocols described in Section 3.3 correlate with rankings of held out task sequences. All protocols used half the available seeds/task sequences to do model selection and evaluation. The generated method rankings were compared against the "held out" rankings generated by looking at the test accuracy of the methods on the other half of the task sequences. Error bars represent 95% empirical confidence intervals.

As an example, with $n = 20$ seeds, each protocol has 10 seeds to perform both model selection and evaluation. Protocol 1 uses the validation accuracy across all 10 seeds to perform model selection and ranks the selected configuration based on the test accuracy of those 10 seeds. The selection and evaluation are done using the same task sequences. Protocol 2 uses the test accuracy of 5 of the seeds to do model selection, and ranks the selected configurations using the test accuracy of the other 5 seeds. The 5 seeds used for selection represent different task sequences than the 5 used for evaluation. Protocol 3 uses the test accuracy of the first 20% of tasks in the sequence across all 10 seeds to do model selection and uses the test accuracy across the rest of the tasks to evaluate the models. The "held out" oracle rankings are created using test performance on the 10 unused seeds. The best configuration is selected and evaluated using all 10 seeds.

This experiment tests the ability of these protocols to evaluate methods in a way that can transfer to different task sequences. Our results (Figure 3) show that while there is no clear winner across

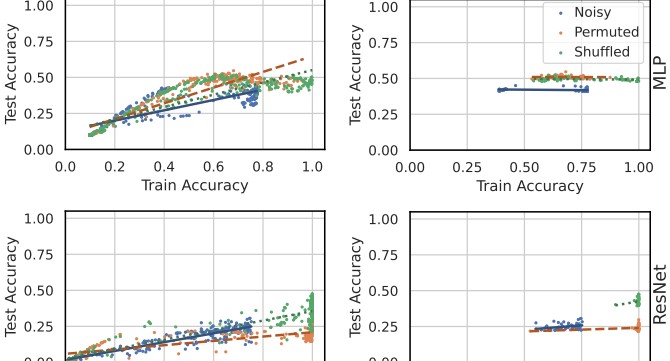

| Shift | Model | Slope | *p*-value |
|---|---|---|---|
| Shuffled | MLP | -0.01 | 0.32 |
| | ResNet | 0.28 | 0.22 |
| Permuted | MLP | -0.0066 | 0.72 |
| | ResNet | 0.05 | 0.22 |
| Noisy | MLP | 0.02 | 0.45 |
| | **ResNet** | **0.12** | **0.011** |

(c) Correlation between train accuracy and test accuracy for top 20% of all configurations. The bolded entry is the only one with a statistically significant positive correlation between train and test accuracy.

(a) Train accuracy vs Test accuracy for all configurations sampled in our study, coded by shift type.

(b) Train accuracy vs Test accuracy for the top 20% of configurations sampled in our study, coded by shift type.

Figure 4: A detailed look at the correlation between train accuracy and test accuracy achieved by different configurations in our study. The top row shows MLP configurations, and the bottom row shows ResNet-18 configurations. When considering all sampled configurations, there is a positive relationship between train and test accuracy. When focusing on only the top 20% (i.e. the configurations that are likely to be selected at the end of a hyperparameter search), however, the relationship between train and test accuracy becomes weak if not nonexistent.

all settings, Protocol 2 is the only one that is not outright beaten by another protocol (taking into account the confidence interval (CI)), and it is the outright best on Shuffled CIFAR-10. Protocol 2 outright beats Protocol 1 on 2 of the 6 settings, and approximately matches performance (within CI) on the other 4, providing evidence for the claim in Cha & Cho (2024) that Protocol 1 overfits compared to Protocol 2. Protocol 3 similarly is outright worse on 2 settings compared to Protocol 2, and within the CI on the other 4. Thus, we argue that there is not much advantage to be gained from using Protocols 1 or 3, and at least a few settings where it is disadvantageous to do so.

## 4.2 THE PERFORMANCE OF CURRENT METHODS

We now evaluate the performance of methods from the literature across both nonstationarities and architectures (Figure 2). Based on the results of the previous section, we use protocol 2 for model selection and evaluation. Figure 2a shows both the training and testing performance of each method on our suite of benchmarks. We observe the following results: (1) A well tuned *Online* baseline does surprisingly well. The only evaluation where it is clearly last place is test accuracy on Permuted CIFAR-100. On the others, it beats at least one and often times multiple baselines that claim to outperform it. (2) Training accuracy starts to saturate on 4 out of the 6 settings we study. For the larger ResNet-18 architecture, nearly all methods are saturated on all the distribution shifts. (3) When looking at test performance, *Hare & Tortoise* and *CReLU* do well across all the settings we examine. *Shrink & Perturb* does well on the ResNet architecture (top 2 for each setting), while *L2 Init* does well with the MLP architecture (top 3 in each setting) and struggles with ResNet (bottom 2 in each setting).

We also look at the correlation between the rankings of the methods on each setting in Figure 2b. We see that with a couple of exceptions, the method rankings on different settings and evaluation criteria (train vs test accuracy) are not strongly correlated with each other. Especially comparing rankings generated from train accuracy and those from test accuracy, we find that many of them are in fact anti-correlated, implying that a method that does better on one does worse on the other.

### 4.3 Correlation between training and testing plasticity

Many previous works exploring plasticity and nonstationary optimization often focus on trainability, arguing that fixing trainability is a precondition to fixing generalizability. Many works do not even report testing accuracy and often use distribution shifts such as Random Label Assignment, where each individual example is assigned a random label, with the goal being to test the limits of trainability and ignoring generalizability.

In this section, we question this line of reasoning. In Figure 4, we see the correlation between train and test accuracy of different configurations. The train and test accuracies were obtained by averaging the results of all seeds for that configuration. In Figure 4a, we see a positive correlation when plotting all configurations, but there is a leveling off of the test accuracy after a certain point. When focusing on just the top 20% of configurations in each setting (Fig. 4b) (i.e. the configurations that are likely to be actually chosen after a hyperparameter search), the positive correlation essentially disappears for most settings. There is in fact a (statistically insignificant) negative correlation for a couple of the MLP settings, and even for the settings with a positive correlation, Noisy CIFAR-100 with ResNet is the only setting where we see a statistically significant result ($p < .05$). Thus, trainability correlates with generalizability *only up to a point*, after which continuing to improve trainability does not end up correlating with the end goal of improving model performance. This suggests that (1) for the types of settings presented in this study (which are representative of what is currently studied in the literature), we should shift our focus from improving trainability to the problem of improving generalizability. (2) Studying trainability could still be a valuable problem, but we should find harder settings to do so.

### 4.4 How many seeds do you need to evaluate a Method?

Many prior works use anywhere between 1 and 5 seeds for hyperparameter selection and then run the selected configuration with more seeds. Figure 5 looks at the effect of the number of seeds used when selecting hyperparameter configurations. For a given number of seeds, we perform statistical bootstrapping with 1000 trials by resampling the seeds used to do model selection. We use the sampled measurements to create rankings over the configurations being evaluated for each method, and compare them to the "oracle" rankings which we compute by taking the average of all available seeds.

Figure 5a shows that to ensure that the absolute best configuration is chosen for every method, you likely need to use almost all the available seeds. If we relax our requirements slightly, however, in Figures 5b and 5c, we see that for most settings (aside from Permuted CIFAR-10 with MLP and Noisy CIFAR-100 with ResNet-18), just a couple of seeds are enough to ensure that the top configuration is actually the oracle best and that the oracle best configuration is evaluated as a top 3 configuration in the selection process.

These results imply that we do not need very many seeds to do effective hyperparameter selection. Furthermore, the results from Figures 5b and 5c point to a potentially effective use case of sequential hyperparameter optimization approaches that can refine configurations or request more resources for promising configurations.

### 4.5 How many tasks do you need to evaluate a Method?

Protocol 3 in Section 3.3 proposes that we only use a subset of the tasks in the sequence to do model selection. We see in Figure 6a, that unfortunately, this protocol is not able to find the best configurations for future tasks for the methods we studied. This suggests that either our methods might not be robust learners that can maintain performance no matter what stage of training they are in and/or we need better ways of selecting the best configuration rather than best average accuracy over tasks seen so far. Creating methods that can succeed in this protocol (or something similar) can help us create agents that do well on lifetimes longer than what the protocol sees in the selection stage, a necessary step in creating lifelong learning agents with unbounded lifetimes.

While not able to maintain performance for just future tasks, if we change our evaluation criteria to accuracy on all tasks (Figure 6b), including those already seen by the learner, we see that for many settings (e.g. Shuffled CIFAR-10, CIFAR-100, Permuted CIFAR-10), we can perfectly select the best configuration with fewer than half the tasks in the full sequence, and can do respectably

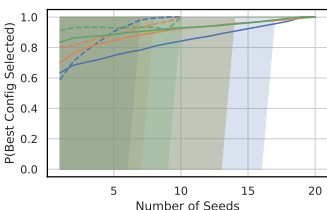 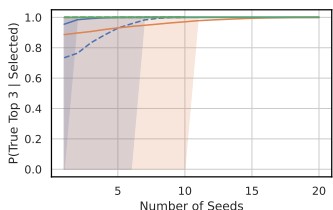 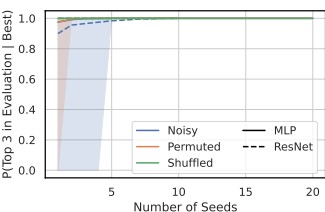

(a) Probability best configuration is selected for any given method.

(b) Probability that the configuration evaluated as best is actually a top 3 configuration.

(c) Probability that the best configuration is evaluated as a top 3 configuration.

Figure 5: A look at how the quality of the configurations selected by a hyperparameter search changes as you vary the number of seeds used to evaluate the configurations in the search. For most settings, to identify the absolute best configuration, you need a large number of seeds, but if allowed to select multiple configurations, even 1 or 2 seeds can be enough. All figures were generated by doing statistical bootstrapping and show the empirical 95% confidence interval. Note, that the lines end at different x-values since the different settings have a different number of total seeds. The "True" ranking mentioned in the figures refers to the ranking generated by using all 20 seeds.

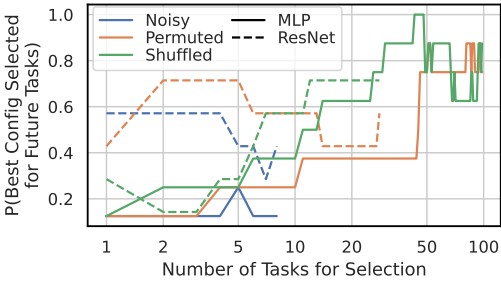 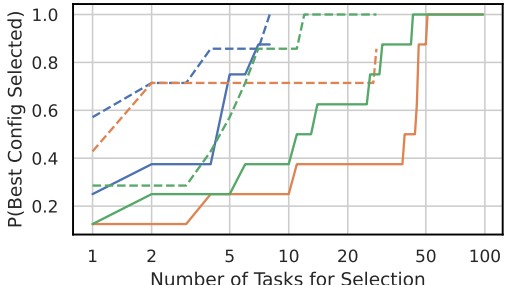

(a) Probability that a configuration selected after $n$ tasks of evaluation will be the best configuration for the rest of the tasks.

(b) Probability that a configuration selected after $n$ tasks of evaluation will be the best configuration when evaluating over all tasks.

Figure 6: Compares the effect of using fewer tasks for hyperparameter selection. If the metric we care about is the average accuracy across all tasks, then a comparably small number of tasks can be used. However, future task performance is more difficult to predict with just prior tasks. Note, the lines end at different x-values since the different settings have different numbers of tasks.

on the other settings as well with fewer tasks. Thus, during hyperparameter optimization, we can likely short circuit a run early and still potentially have a good estimate for the configuration's performance.

### 4.6 How many Hyperparameter Configurations do you need to evaluate?

Here, we look at the benefits of sampling more configurations on the quality of the configuration selected. In Figure 7, we see that across all the settings, sampling more configurations helps up to a certain point, with diminishing returns. After 20-30 configurations, the expected improvement with each additional config for most settings becomes negligible. This is not even considering the fact that our searcher was an unintelligent random searcher. More intelligent algorithms can potentially be even more efficient with the configuration budget.

## 5 Discussion and Takeaways

With the growing interest and body of work in nonstationary optimization, it's important to ensure that we design our experiments in a principled manner, such that our results are significant, reproducible, and can be compared to by future work. We find that under similar hyperparameter search

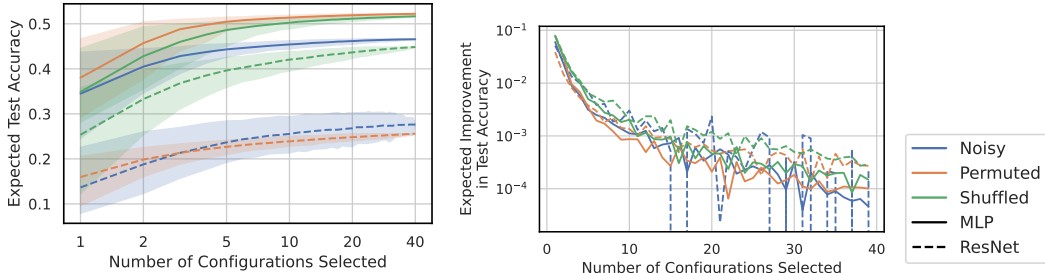

(a) Expected test accuracy vs the number of configurations sampled in the hyperparameter search. Shaded regions are 95% confidence intervals.

(b) Expected improvement with each additional configuration sampled in the hyperparameter search.

Figure 7: There are diminishing returns with sampling more configurations in the search. Even with a random searcher, performance improvement stalls after about 20-30 configurations. A smarter searcher can potentially be even more efficient.

protocols, there is no method that clearly outperforms all others across different types of distribution shifts and architectures. Our paper also examines a host of design decisions that go into designing these experiments in the context of nonstationary optimization, to enable that future work.

**Hyperparameter Search Protocol**    We should not do model selection on the same task sequences that we use to report our evaluation on as that can lead to performance overestimation and method rankings that do not transfer to different task sequences. Furthermore, our current methods, do not perform well in the case where the first few tasks are used to do model selection for the rest of the model's "lifetime", as proposed in (Mesbahi et al., 2024). The ability to select hyperparameters that transfer to timescales not seen in the hyperparameter selection stage is essential to create lifelong agents with unbounded lifetimes. The failure of current methods/protocols to do so invites further research.

**Training vs Testing Plasticity**    The plasticity community has focused much of its effort on creating methods to mitigate the loss of training accuracy, as a prerequisite to eventually improving performance on generalization accuracy. For the types of datasets currently studied by the community, this approach is unsound, as improvements in the ability to maintain training accuracy do not lead to improvements in the ability to maintain generalizability. We should be studying trainability on harder settings where improvements in trainability lead to improvements in generalizability.

**Creating Resource Efficient Hyperparameter Searches**    We explore the effect of three factors that can affect the resources used in a nonstationary optimization hyperparameter search: number of seeds, number of tasks used in the search, and number of configurations sampled. For many settings, you do not need a large number of seeds while doing a hyperparameter search. When using a few seeds, a viable approach could also be to select multiple configurations and train them with more seeds to get a better estimate before selecting a final configuration. We also find that in many settings, you can reduce the number of tasks by as much as 50% and still be able to identify the best configuration for the full task sequence. Finally, even when doing an unintelligent random search, you do not need to sample more than 20-30 configurations on most settings to find performant configurations.

We specifically designed our study around datasets, distribution shifts, and architectures that have been used by several prior works in this field to ensure that our findings are directly useful to the community. We hope that these findings will also enable good research for researchers with access to fewer resources, as our work points to several ways to make experiments more efficient.

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

| Method | Parameter | Values |
|---|---|---|
| Base | L2 Weight | $0.0, 0.01, 0.0001$ |
| | Optimizer | SGD, Adam |
| | Learning Rate | $0.1, 0.01, 0.001, 0.0001, 0.00001$ |
| | $\beta_1$ | $0.9, 0.0$ |
| | $\beta_2$ | $0.99, 0.999, 0.9999$ |
| | $\epsilon$ | $1 \times 10^{-4}, 1 \times 10^{-8}, 1 \times 10^{-16}$ |
| Hare & Tortoise | Base Optimizer | SGD, Adam |
| | Reset Period | $200, 400, 1000, 2000, 4000, 20000$ |
| | HT $\mu$ | $0.98, 0.99, 0.995, 0.999, 0.9995, 0.9999$ |
| L2 Init | Regularization Weight | $0.1, 0.01, 0.001, 0.0001$ |
| | L2 Weight | $0.0$ |
| ReDo | Reset Period | $200, 400, 1000, 2000, 4000, 20000$ |
| | Dormancy Threshold | $0.01, 0.02, 0.05, 0.1, 0.2, 0.5$ |
| Shrink & Perturb | Noise Scale | $0.001, 0.01, 0.1, 1.0$ |
| | Shrink Weight | $0.0, 0.2, 0.4, 0.6, 0.8, 1.0$ |
| | L2 Weight | $0.0$ |
| CBP | Decay Rate | $0.9, 0.99, 0.999$ |
| | Maturity Threshold | $100, 1000, 10000$ |
| | Replacement Rate | $1 \times 10^{-3}, 1 \times 10^{-4}, 1 \times 10^{-5}, 1 \times 10^{-6}$ |

Table 1: The search space used for every method in our study. Every method included the Base space as part of its search, and some methods added additional hyperparameters.

## A    HYPERPARAMETER SEARCH SPACES

We present the search space used for each method in Table 1.

## B    TRAINING DETAILS

We briefly describe the training procedure we used for training. All experiments used batch size 256. For the MLP experiments, each task consisted of 100,000 gradient steps of training with batch size 256. For ResNet-18, each task consisted of 20,000 gradient steps also with batch size 256. Each seed ran a different randomly generated task sequence. All experiments were run in JAX (Bradbury et al., 2018), parallelized over seeds.

We also list the hyperparameter configuration for the best sampled configuration for each setting and method in Table 2.

## C    HYPERPARAMETER IMPORTANCE

We now look at the importance of different hyperparameters across different methods and settings (Figure 8). We calculate the PED-ANOVA (Watanabe et al., 2023) importance score for each hyperparameter, which describes the relative importance of each hyperparameter in predicting final performance. The learning rate and L2 Loss weight value seem to be consistently important across all methods. Method specific hyperparameters tend to be fairly important.

| Setting | Model | Method | Test Accuracy | Hyperparameters |
|---|---|---|---|---|
| Noisy CIFAR-10 | MLP | CBP | 0.46 | Optimizer=SGD,LR=1.00e-04,L2 Loss Weight=1.00e-04,Adam $\epsilon$=1.00e-08,Adam $\beta_1$=9.00e-01, Adam $\beta_2$=9.90e-01,Replacement Rate=1.00e-05,Maturity Threshold=1.00e+02,Decay Rate=9.90e-01, |
| Noisy CIFAR-10 | MLP | CReLU | 0.48 | Optimizer=SGD,LR=1.00e-04,L2 Loss Weight=1.00e-04,Adam $\epsilon$=1.00e-08,Adam $\beta_1$=9.00e-01, Adam $\beta_2$=9.90e-01 |
| Noisy CIFAR-10 | MLP | H&T | 0.48 | LR=1.00e-05,L2 Loss Weight=1.00e-04,Adam $\epsilon$=1.00e-08,Adam $\beta_1$=9.00e-01,Adam $\beta_2$=9.99e-01, Base Opt=Adam,Mom=1.00e+00,Reset Period=1.00e+03 |
| Noisy CIFAR-10 | MLP | L2 Init | 0.47 | Optimizer=SGD,LR=1.00e-04,Adam $\epsilon$=1.00e-08,Adam $\beta_1$=9.00e-01,Adam $\beta_2$=9.90e-01, L2 Init Weight=1.00e-04 |
| Noisy CIFAR-10 | MLP | LN | 0.44 | Optimizer=SGD,LR=1.00e-05,L2 Loss Weight=1.00e-04,Adam $\epsilon$=1.00e-08,Adam $\beta_1$=9.00e-01, Adam $\beta_2$=9.90e-01 |
| Noisy CIFAR-10 | MLP | Online | 0.47 | Optimizer=SGD,LR=1.00e-04,L2 Loss Weight=1.00e-04,Adam $\epsilon$=1.00e-08,Adam $\beta_1$=9.00e-01, Adam $\beta_2$=9.90e-01 |
| Noisy CIFAR-10 | MLP | Redo | 0.47 | Optimizer=SGD,LR=1.00e-04,L2 Loss Weight=1.00e-02,Adam $\epsilon$=1.00e-08,Adam $\beta_1$=9.00e-01, Adam $\beta_2$=9.90e-01,Reset Period=4.00e+02,Dormancy Threshold=2.00e-02 |
| Noisy CIFAR-10 | MLP | S&P | 0.46 | Optimizer=SGD,LR=1.00e-02,Adam $\epsilon$=1.00e-08,Adam $\beta_1$=9.00e-01,Adam $\beta_2$=9.90e-01, Shrink Weight=4.00e-01,Noise Scale=1.00e-03 |
| Noisy CIFAR-100 | ResNet-18 | CBP | 0.30 | Optimizer=Adam,LR=1.00e-04,L2 Loss Weight=1.00e-04,Adam $\epsilon$=1.00e-16,Adam $\beta_1$=0.00e+00, Adam $\beta_2$=9.99e-01,Replacement Rate=1.00e-06,Maturity Threshold=1.00e+04,Decay Rate=9.99e-01, |
| Noisy CIFAR-100 | ResNet-18 | CReLU | 0.35 | Optimizer=Adam,LR=1.00e-03,L2 Loss Weight=0.00e+00,Adam $\epsilon$=1.00e-16,Adam $\beta_1$=0.00e+00, Adam $\beta_2$=9.90e-01 |
| Noisy CIFAR-100 | ResNet-18 | H&T | 0.36 | LR=1.00e-03,L2 Loss Weight=1.00e-04,Adam $\epsilon$=1.00e-04,Adam $\beta_1$=0.00e+00,Adam $\beta_2$=9.90e-01, Base Opt=Adam,Mom=9.99e-01,Reset Period=2.00e+02 |
| Noisy CIFAR-100 | ResNet-18 | L2 Init | 0.26 | Optimizer=Adam,LR=1.00e-05,Adam $\epsilon$=1.00e-16,Adam $\beta_1$=9.00e-01,Adam $\beta_2$=9.99e-01, L2 Init Weight=1.00e-04 |
| Noisy CIFAR-100 | ResNet-18 | Online | 0.30 | Optimizer=Adam,LR=1.00e-04,L2 Loss Weight=0.00e+00,Adam $\epsilon$=1.00e-04,Adam $\beta_1$=0.00e+00, Adam $\beta_2$=9.99e-01 |
| Noisy CIFAR-100 | ResNet-18 | Redo | 0.31 | Optimizer=Adam,LR=1.00e-03,L2 Loss Weight=0.00e+00,Adam $\epsilon$=1.00e-04,Adam $\beta_1$=9.00e-01, Adam $\beta_2$=9.99e-01,Reset Period=4.00e+02,Dormancy Threshold=2.00e-02 |
| Noisy CIFAR-100 | ResNet-18 | S&P | 0.35 | Optimizer=Adam,LR=1.00e-04,Adam $\epsilon$=1.00e-04,Adam $\beta_1$=9.00e-01,Adam $\beta_2$=9.90e-01, Shrink Weight=4.00e-01,Noise Scale=1.00e-03 |
| Permuted CIFAR-10 | MLP | CBP | 0.52 | Optimizer=SGD,LR=1.00e-03,L2 Loss Weight=1.00e-02,Adam $\epsilon$=1.00e-08,Adam $\beta_1$=9.00e-01, Adam $\beta_2$=9.90e-01,Replacement Rate=1.00e-04,Maturity Threshold=1.00e+02,Decay Rate=9.00e-01, |
| Permuted CIFAR-10 | MLP | CReLU | 0.52 | Optimizer=Adam,LR=1.00e-05,L2 Loss Weight=1.00e-02,Adam $\epsilon$=1.00e-04,Adam $\beta_1$=9.00e-01, Adam $\beta_2$=1.00e+00 |
| Permuted CIFAR-10 | MLP | H&T | 0.55 | LR=1.00e-04,L2 Loss Weight=1.00e-02,Adam $\epsilon$=1.00e-08,Adam $\beta_1$=9.00e-01,Adam $\beta_2$=1.00e+00, Base Opt=Adam,Mom=9.99e-01,Reset Period=2.00e+02 |
| Permuted CIFAR-10 | MLP | L2 Init | 0.53 | Optimizer=SGD,LR=1.00e-02,Adam $\epsilon$=1.00e-08,Adam $\beta_1$=9.00e-01,Adam $\beta_2$=9.90e-01, L2 Init Weight=1.00e-02 |
| Permuted CIFAR-10 | MLP | LN | 0.51 | Optimizer=SGD,LR=1.00e-04,L2 Loss Weight=0.00e+00,Adam $\epsilon$=1.00e-08,Adam $\beta_1$=9.00e-01, Adam $\beta_2$=9.90e-01 |
| Permuted CIFAR-10 | MLP | Online | 0.51 | Optimizer=Adam,LR=1.00e-05,L2 Loss Weight=1.00e-04,Adam $\epsilon$=1.00e-04,Adam $\beta_1$=9.00e-01, Adam $\beta_2$=9.90e-01 |
| Permuted CIFAR-10 | MLP | Redo | 0.51 | Optimizer=SGD,LR=1.00e-03,L2 Loss Weight=1.00e-04,Adam $\epsilon$=1.00e-08,Adam $\beta_1$=9.00e-01, Adam $\beta_2$=9.90e-01,Reset Period=4.00e+03,Dormancy Threshold=2.00e-01 |
| Permuted CIFAR-10 | MLP | S&P | 0.53 | Optimizer=SGD,LR=1.00e-03,Adam $\epsilon$=1.00e-08,Adam $\beta_1$=9.00e-01,Adam $\beta_2$=9.90e-01, Shrink Weight=2.00e-01,Noise Scale=1.00e-03 |
| Permuted CIFAR-100 | ResNet-18 | CBP | 0.24 | Optimizer=SGD,LR=1.00e-01,L2 Loss Weight=1.00e-02,Adam $\epsilon$=1.00e-08,Adam $\beta_1$=9.00e-01, Adam $\beta_2$=9.90e-01,Replacement Rate=1.00e-03,Maturity Threshold=1.00e+03,Decay Rate=9.90e-01, |
| Permuted CIFAR-100 | ResNet-18 | CReLU | 0.30 | Optimizer=SGD,LR=1.00e-01,L2 Loss Weight=1.00e-02,Adam $\epsilon$=1.00e-08,Adam $\beta_1$=9.00e-01, Adam $\beta_2$=9.90e-01 |
| Permuted CIFAR-100 | ResNet-18 | H&T | 0.26 | LR=1.00e-01,L2 Loss Weight=1.00e-02,Adam $\epsilon$=1.00e-08,Adam $\beta_1$=9.00e-01,Adam $\beta_2$=9.90e-01, Base Opt=SGD,Mom=1.00e+00,Reset Period=1.00e+04 |
| Permuted CIFAR-100 | ResNet-18 | L2 Init | 0.24 | Optimizer=SGD,LR=1.00e-01,Adam $\epsilon$=1.00e-08,Adam $\beta_1$=9.00e-01,Adam $\beta_2$=9.90e-01, L2 Init Weight=1.00e-03 |
| Permuted CIFAR-100 | ResNet-18 | Online | 0.23 | Optimizer=SGD,LR=1.00e-02,L2 Loss Weight=1.00e-02,Adam $\epsilon$=1.00e-08,Adam $\beta_1$=9.00e-01, Adam $\beta_2$=9.90e-01 |
| Permuted CIFAR-100 | ResNet-18 | Redo | 0.26 | Optimizer=SGD,LR=1.00e-01,L2 Loss Weight=1.00e-02,Adam $\epsilon$=1.00e-08,Adam $\beta_1$=9.00e-01, Adam $\beta_2$=9.90e-01,Reset Period=2.00e+02,Dormancy Threshold=5.00e-01 |
| Permuted CIFAR-100 | ResNet-18 | S&P | 0.28 | Optimizer=SGD,LR=1.00e-01,Adam $\epsilon$=1.00e-08,Adam $\beta_1$=9.00e-01,Adam $\beta_2$=9.90e-01, Shrink Weight=2.00e-01,Noise Scale=1.00e-01 |
| Shuffled CIFAR-10 | MLP | CBP | 0.51 | Optimizer=Adam,LR=1.00e-05,L2 Loss Weight=1.00e-02,Adam $\epsilon$=1.00e-08,Adam $\beta_1$=0.00e+00, Adam $\beta_2$=9.99e-01,Replacement Rate=1.00e-03,Maturity Threshold=1.00e+03,Decay Rate=9.99e-01, |
| Shuffled CIFAR-10 | MLP | CReLU | 0.52 | Optimizer=Adam,LR=1.00e-03,L2 Loss Weight=1.00e-02,Adam $\epsilon$=1.00e-04,Adam $\beta_1$=9.00e-01, Adam $\beta_2$=1.00e+00 |
| Shuffled CIFAR-10 | MLP | H&T | 0.51 | LR=1.00e-05,L2 Loss Weight=1.00e-02,Adam $\epsilon$=1.00e-16,Adam $\beta_1$=0.00e+00,Adam $\beta_2$=1.00e+00, Base Opt=Adam,Mom=1.00e+00,Reset Period=4.00e+03 |
| Shuffled CIFAR-10 | MLP | L2 Init | 0.53 | Optimizer=SGD,LR=1.00e-02,Adam $\epsilon$=1.00e-08,Adam $\beta_1$=9.00e-01,Adam $\beta_2$=9.90e-01, L2 Init Weight=1.00e-02 |
| Shuffled CIFAR-10 | MLP | LN | 0.51 | Optimizer=Adam,LR=1.00e-03,L2 Loss Weight=1.00e-02,Adam $\epsilon$=1.00e-08,Adam $\beta_1$=0.00e+00, Adam $\beta_2$=9.90e-01 |
| Shuffled CIFAR-10 | MLP | Online | 0.51 | Optimizer=Adam,LR=1.00e-03,L2 Loss Weight=1.00e-02,Adam $\epsilon$=1.00e-16,Adam $\beta_1$=0.00e+00, Adam $\beta_2$=9.90e-01 |
| Shuffled CIFAR-10 | MLP | Redo | 0.52 | Optimizer=Adam,LR=1.00e-03,L2 Loss Weight=1.00e-02,Adam $\epsilon$=1.00e-16,Adam $\beta_1$=0.00e+00, Adam $\beta_2$=9.99e-01,Reset Period=4.00e+03,Dormancy Threshold=5.00e-01 |
| Shuffled CIFAR-10 | MLP | S&P | 0.52 | Optimizer=SGD,LR=1.00e-02,Adam $\epsilon$=1.00e-08,Adam $\beta_1$=9.00e-01,Adam $\beta_2$=9.90e-01, Shrink Weight=8.00e-01,Noise Scale=1.00e-01 |
| Shuffled CIFAR-100 | ResNet-18 | CBP | 0.45 | Optimizer=SGD,LR=1.00e-01,L2 Loss Weight=1.00e-02,Adam $\epsilon$=1.00e-08,Adam $\beta_1$=9.00e-01, Adam $\beta_2$=9.90e-01,Replacement Rate=1.00e-06,Maturity Threshold=1.00e+02,Decay Rate=9.99e-01, |
| Shuffled CIFAR-100 | ResNet-18 | CReLU | 0.47 | Optimizer=SGD,LR=1.00e-01,L2 Loss Weight=1.00e-02,Adam $\epsilon$=1.00e-08,Adam $\beta_1$=9.00e-01, Adam $\beta_2$=9.90e-01 |
| Shuffled CIFAR-100 | ResNet-18 | H&T | 0.46 | LR=1.00e-02,L2 Loss Weight=1.00e-04,Adam $\epsilon$=1.00e-08,Adam $\beta_1$=9.00e-01,Adam $\beta_2$=9.90e-01, Base Opt=Adam,Mom=1.00e+00,Reset Period=4.00e+02 |
| Shuffled CIFAR-100 | ResNet-18 | L2 Init | 0.39 | Optimizer=SGD,LR=1.00e-01,Adam $\epsilon$=1.00e-08,Adam $\beta_1$=9.00e-01,Adam $\beta_2$=9.90e-01, L2 Init Weight=1.00e-03 |
| Shuffled CIFAR-100 | ResNet-18 | Online | 0.45 | Optimizer=SGD,LR=1.00e-01,L2 Loss Weight=1.00e-02,Adam $\epsilon$=1.00e-08,Adam $\beta_1$=9.00e-01, Adam $\beta_2$=9.90e-01 |
| Shuffled CIFAR-100 | ResNet-18 | Redo | 0.45 | Optimizer=SGD,LR=1.00e-01,L2 Loss Weight=1.00e-02,Adam $\epsilon$=1.00e-08,Adam $\beta_1$=9.00e-01, Adam $\beta_2$=9.90e-01,Reset Period=1.00e+04,Dormancy Threshold=1.00e-01 |
| Shuffled CIFAR-100 | ResNet-18 | S&P | 0.48 | Optimizer=Adam,LR=1.00e-03,Adam $\epsilon$=1.00e-08,Adam $\beta_1$=9.00e-01,Adam $\beta_2$=1.00e+00, Shrink Weight=8.00e-01,Noise Scale=1.00e-01 |

Table 2: List of every setting, the best configuration on that setting, and the test accuracy for that setting.

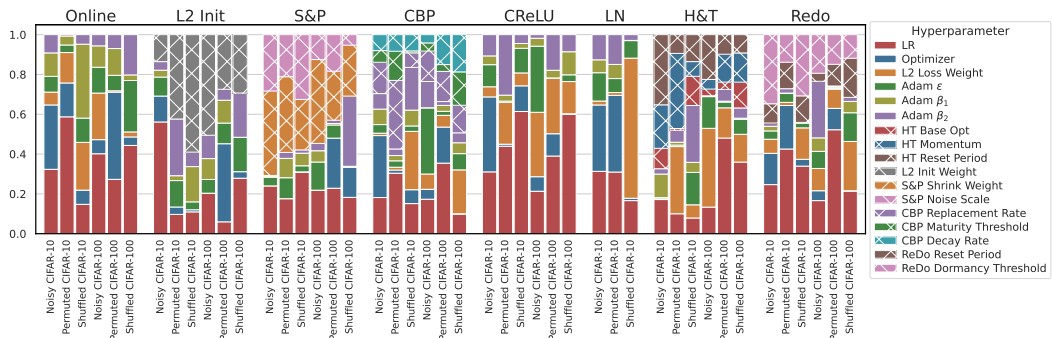

Figure 8: PED-ANOVA (Watanabe et al., 2023) hyperparameter importance for each hyperparameter in our search. Hatched bars are method specific hyperparameters.

# D  ADDITIONAL RESULTS

## D.1  ADDITIONAL DISCUSSION ON SEEDS

In Figure 9a, we look at the probability that the overall method rankings change depending on the number of seeds used. The ranking is fairly stable for most of the settings, and Noisy CIFAR-100, Shuffled CIFAR-10, and Permuted CIFAR-10 are the only settings that required more than 5 seeds to get a fully stable ranking. Figure 9b shows the correlation between the sampled rankings and the oracle ranking, and it shows that the noisy settings tend to have lower correlation and need more seeds. Finally, 9c shows that there is only a modest improvement in expected average test accuracy over methods (i.e. the expected value of the average of the test accuracies of the selected configurations for each method).

In Figures 10 and 11, we present a non aggregated (by method) version of the results from Figures 5 and 9 to examine the effect of the number of seeds on hyperparameter searches for each specific method. While the results vary from method to method, a few trends do emerge. We do not need very many seeds to select the best *Hare & Tortoise* configuration. Figure 10b shows that other than the *Online* baseline on Noisy CIFAR-100 with ResNet, the configurations selected as the best configuration in the search is very likely to at least be a top 3 configuration even with 1 seed. From Figure 10c, we see that the best configuration will be evaluated as a top 3 configuration in the search with very few seeds for every case except Noisy CIFAR-100 and *Online* on Permuted CIFAR-10. Figures 11a and 11b show approximately the ordering of how sensitive the configuration rankings are for different methods to the number of seeds used in model selection. Although there is a lot of overlap, you can still see some separation between methods. Figure 11c shows the approximate range expected for the test accuracy of a method given a certain number of seeds used for selection. For most methods on most settings, there is not a very large spread even with a small number of seeds. For a few methods, however, there is a large spread when using small numbers of seeds which narrows at higher numbers of seeds.

## D.2  SAMPLING EXTRA CONFIGURATIONS

In Figure 12, we examine the effect of sampling extra configurations separately for each method across all settings. We see that there is not much difference between methods for the expected improvement per extra configuration or the width of the range of expected final test accuracy.

## D.3  EVALUATION PROTOCOLS

We dive deeper into the protocols presented in Section 3.3 and the results presented in Section 4.1 in Figure 13. We see that Protocol 2 is still superior in the ability to transfer to held out data, matching the held out ranking exactly more often than Protocol 1 or Protocol 3.

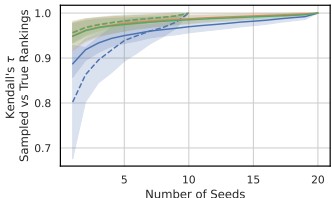 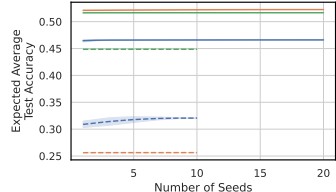

(a) Probability that the method ranking matches the true ranking when using $n$ seeds to select the best configuration for each method.

(b) Kendall's $\tau$ between the true rankings and the rankings created using only $n$ seeds.

(c) The expected best test accuracy averaged across all methods given a certain number of seeds.

Figure 9: A look at how the quality of the configurations selected by a hyperparameter search changes as you vary the number of seeds used to evaluate the configurations in the search. All figures were generated by doing statistical bootstrapping and show the empirical 95% confidence interval. Note, that the lines end at different x-values since the different settings have a different number of total seeds. The "True" ranking mentioned in the figures refers to the ranking generated by using all 20 seeds.

## D.4  METHOD RANKINGS

In Figure 14, we see how the method rankings change depending on the architecture, the distribution shift, and whether we are optimizing for train or test accuracy. We see that there is not a method that consistently dominates across all settings. *Hare & Tortoise* does well on test accuracy, but not well on train accuracy. *L2 Init* does well on MLP test accuracy, but badly on ResNet test accuracy. A method such as LayerNorm which does badly on test accuracy performs well on train accuracy.

## D.5  TRAINING VS TESTING PLASTICITY

In Figure 15, we see that the relationship between train and test loss of the different configurations in our study. A decrease in train loss correlates with a decrease in test loss up to a certain point, after which there is a lot of overfitting to the train set. When we focus on just the top 20% of configurations on test loss, we see that there is little to no relationship between train and test loss for most settings (other than Shuffled CIFAR-10).

Figure 16 and Table 3 show the relationship between train and test accuracy for the various different methods in our study with the results separated by setting. In Figure 16a, we see a similar trend as 4a, where train accuracy is positively correlated with test accuracy. When focusing on just the top 20% of configurations of each method (Fig. 16b, Tab. 3), however, there isn't a statistically significant positive correlation between train and test accuracy. In fact, the only place where such a relationship exists is for ReDo with ResNets on Noisy CIFAR-100 (the other statistically significant positive correlations in the table are essentially vertical lines that are not well defined correlations).

## D.6  *CReLU* ADJUSTED FOR NUMBER OF PARAMETERS

The *CReLU* activation function changes the architecture by doubling the size of the activation output. This also increases the number of parameters in the network since for the intermediate layers, the input size is doubled compared to a non-*CReLU* activation.

To adjust for the number of parameters, we ran a search with a smaller architecture such that the number of parameters in the network approximately matches an architecture without *CReLU*. For the MLP architecture, this meant a reduction of the hidden size from 128 to 120, and for the ResNet architecture, we reduced the number of filters in each convolutional layer from 64 to 48. We ran a slightly shortened search with 30 configurations for each setting, and show the results in Figure 17. We can see that the MLP results are approximately the same, with a slight decrease in performance in Permuted CIFAR-10 training accuracy and Shuffled CIFAR-10 test accuracy. For ResNet, the training performance is matched by the smaller network, but the test performance is significantly lower across all settings.

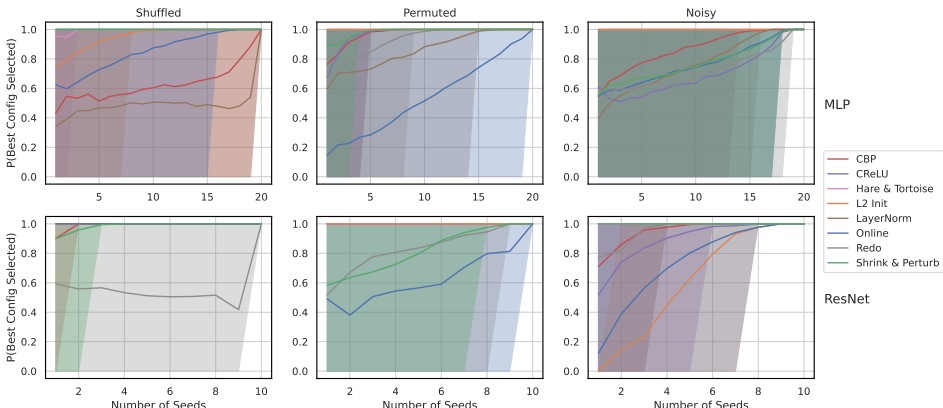

(a) Probability that the best configuration is selected in a hyperparameter search with $n$ seeds for each method.

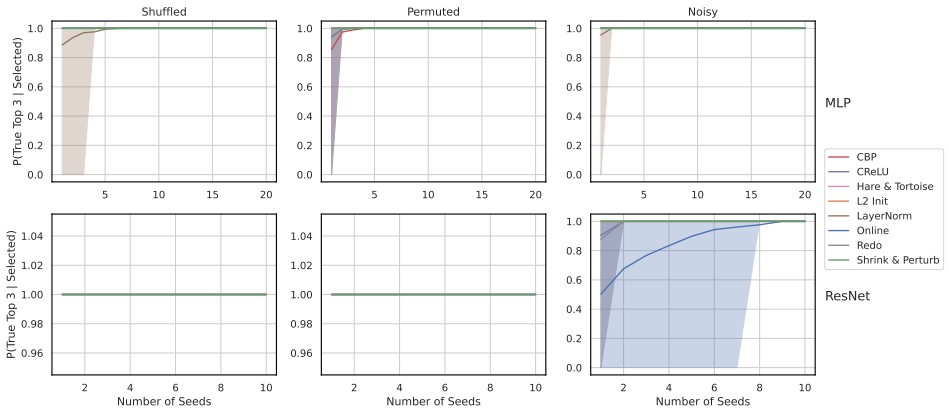

(b) Probability that the configuration evaluated as best is actually a top 3 configuration for each method.

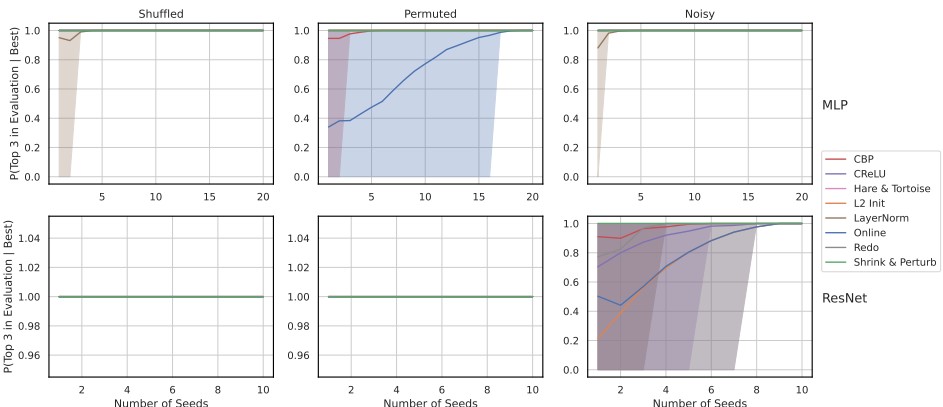

(c) Probability that the best configuration is evaluated as a top 3 configuration for each method.

Figure 10: A version of the plots in Figure 5 analyzing the effect of seeds in hyperparameter searches separated by method. The "True" ranking mentioned in the figures refers to the ranking generated by using all 20 seeds.

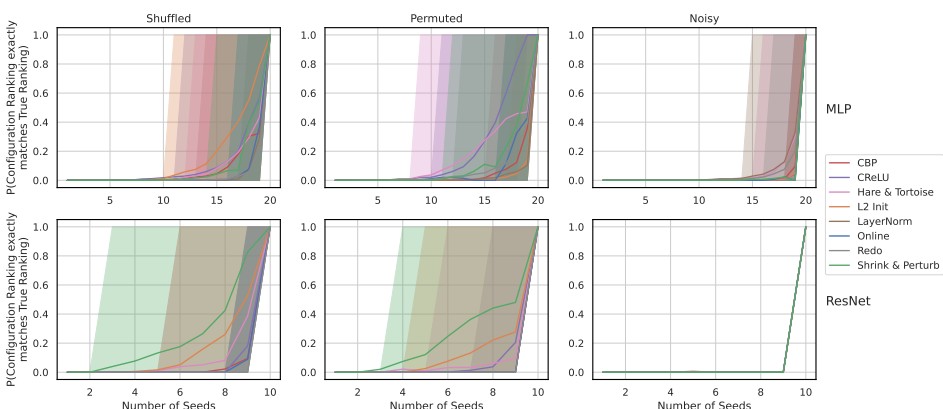

(a) Probability that the ranking for all 40 configurations matches the true ranking for each method.

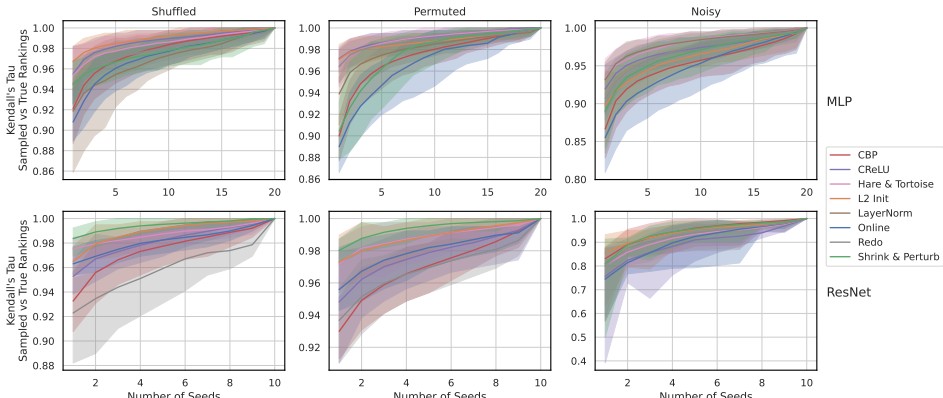

(b) Kendall's $\tau$ between the true configuration rankings and the configuration rankings created using only $n$ seeds for each method.

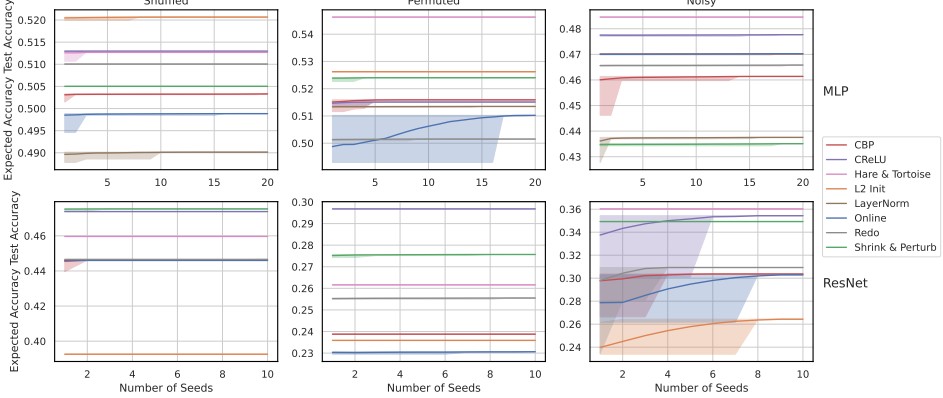

(c) The expected best test accuracy for each method given a certain number of seeds.

Figure 11: A version of the plots in Figure 9 analyzing the effect of seeds in hyperparameter searches separated by method. The "True" ranking mentioned in the figures refers to the ranking generated by using all 20 seeds.

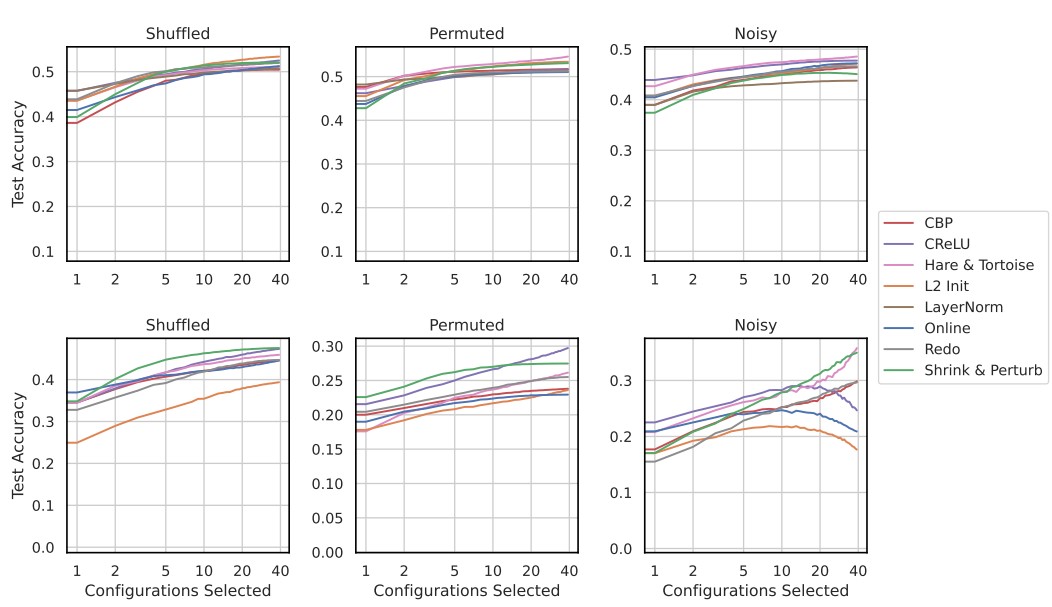

(a) Expected test accuracy per extra configuration sampled for each method across all settings. Top row is MLP, bottom row is ResNet-18.

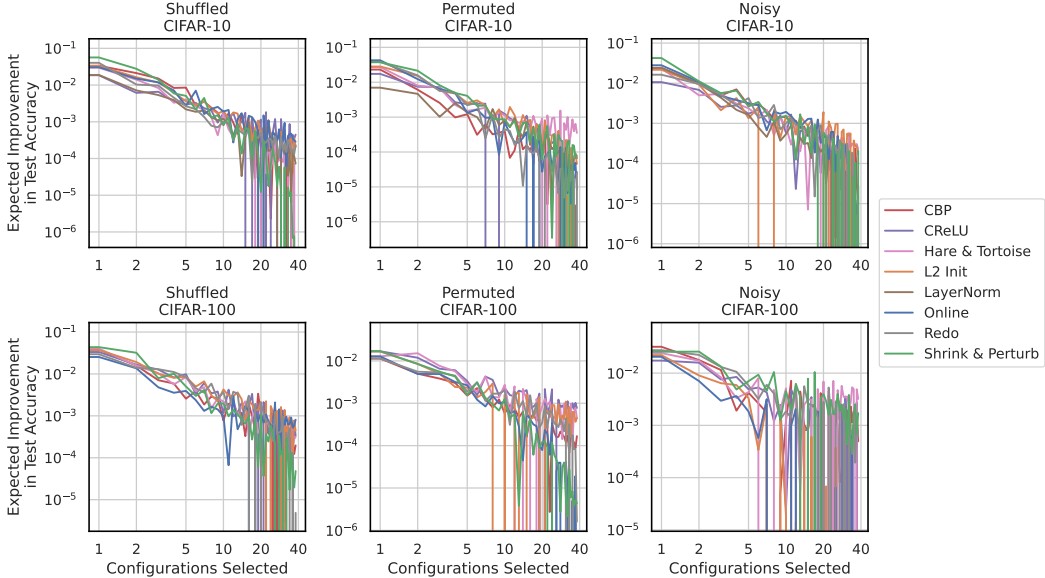

(b) Expected improvement in test accuracy for every extra configuration sampled for each method across all settings. Top row is MLP, bottom row is ResNet-18.

Figure 12: Effect of sampling extra configurations in a hyperparameter search, broken down by method.

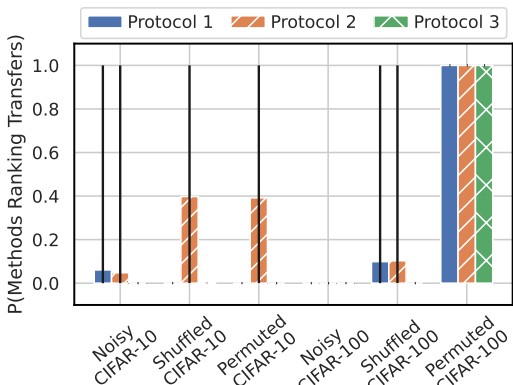

Figure 13: Probability that the ranking generated by the protocol exactly matches the held out ranking.

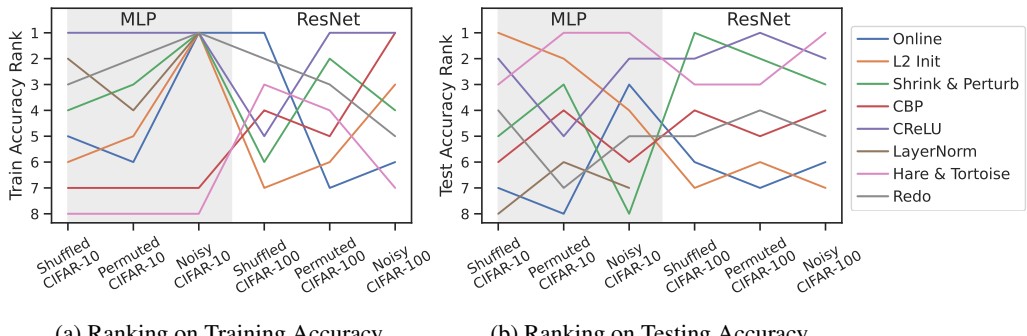

(a) Ranking on Training Accuracy.    (b) Ranking on Testing Accuracy.

Figure 14: The overall performance of different plasticity preserving methods from the literature across different distribution shifts and architectures. There is not a method that consistently dominates all other methods across all settings, but there are some methods that are dominated. Furthermore, ranking on training accuracy does not seem to correlate to ranking on test accuracy.

| Shift | Model | Method | Slope | $p$-value |
|---|---|---|---|---|
| Shuffled | MLP | Online | 0.017 | 0.865 |
| | | L2 Init | -0.131 | 0.060 |
| | | Shrink & Perturb | -0.002 | 0.868 |
| | | CBP | -0.152 | 0.066 |
| | | CReLU | -0.267 | 0.138 |
| | | LayerNorm | 0.010 | 0.821 |
| | | Hare & Tortoise | -0.129 | 0.126 |
| | | Redo | -0.051 | 0.001 |
| | ResNet | Online | 1.650 | 0.454 |
| | | L2 Init | -0.102 | 0.619 |
| | | **Shrink & Perturb** | **6.608** | **0.046** |
| | | CBP | 4.866 | 0.317 |
| | | **CReLU** | **12.757** | **0.013** |
| | | Hare & Tortoise | 0.258 | 0.367 |
| | | Redo | -0.486 | 0.943 |
| Permuted | MLP | Online | -0.030 | 0.431 |
| | | L2 Init | -0.145 | 0.001 |
| | | Shrink & Perturb | 0.007 | 0.866 |
| | | CBP | -0.094 | 0.026 |
| | | CReLU | 0.068 | 0.086 |
| | | LayerNorm | -0.536 | 0.073 |
| | | Hare & Tortoise | 0.027 | 0.714 |
| | | Redo | 0.021 | 0.350 |
| | ResNet | Online | 0.043 | 0.207 |
| | | L2 Init | 0.016 | 0.334 |
| | | Shrink & Perturb | 3.600 | 0.083 |
| | | **CBP** | **5.163** | **0.001** |
| | | CReLU | 1.577 | 0.248 |
| | | Hare & Tortoise | 0.028 | 0.476 |
| | | Redo | 9.888 | 0.205 |
| Noisy | MLP | Online | -0.041 | 0.095 |
| | | L2 Init | -0.027 | 0.020 |
| | | Shrink & Perturb | 0.136 | 0.577 |
| | | CBP | -0.001 | 0.936 |
| | | CReLU | 0.012 | 0.477 |
| | | **LayerNorm** | **1.070** | **0.032** |
| | | Hare & Tortoise | 0.038 | 0.139 |
| | | Redo | 0.003 | 0.883 |
| | ResNet | Online | -0.086 | 0.550 |
| | | L2 Init | -0.045 | 0.191 |
| | | Shrink & Perturb | -0.106 | 0.237 |
| | | CBP | 0.158 | 0.162 |
| | | CReLU | 0.452 | 0.173 |
| | | Hare & Tortoise | 0.091 | 0.636 |
| | | **Redo** | **0.329** | **0.012** |

Table 3: Correlation between train accuracy and test accuracy for top 20% of all configurations.

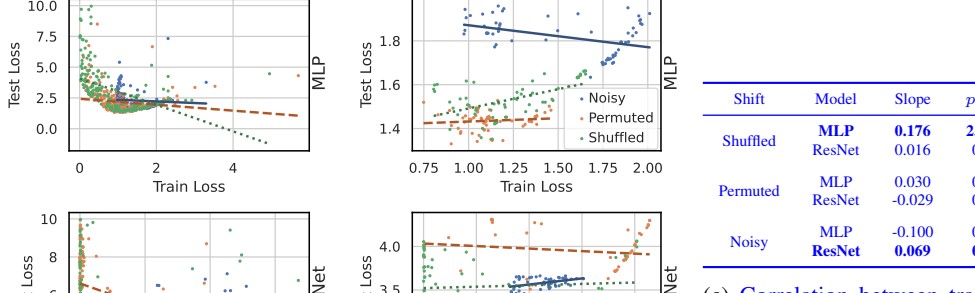

| Shift | Model | Slope | $p$-value |
|---|---|---|---|
| Shuffled | **MLP** | **0.176** | **2.90e-6** |
| | ResNet | 0.016 | 0.690 |
| Permuted | MLP | 0.030 | 0.364 |
| | ResNet | -0.029 | 0.307 |
| Noisy | MLP | -0.100 | 0.001 |
| | **ResNet** | **0.069** | **0.002** |

(c) Correlation between train loss and test loss for top 20% of all configurations. Settings with statistically significant positive correlation are bolded.

(a) Train Loss vs Test Loss for all configurations sampled in our study, coded by shift type. Diverged configurations are filtered out.

(b) Train loss vs Test loss for the top 20% of configurations sampled in our study, coded by shift type.

Figure 15: A detailed look at the correlation between train loss and test loss achieved by different configurations in our study. The top row shows MLP configurations, and the bottom row shows ResNet-18 configurations. Up to a certain point, it does make sense to focus on preserving train loss, but for the best configurations, preserving train loss leads to little to no improvement in test loss.

# E    BOOTSTRAPPING PROCEDURE

With statistical bootstrapping, we sample $B$ datasets of size $n$, compute some statistic using each of these datasets, and use the empirical distribution over the statistic to create confidence intervals or compute standard errors for the sample mean of the statistic (Hastie et al., 2009). In our case, we are sampling partitions over the seeds, $P$, and using the partition to estimate some statistics that are a function of this partition, $f(P)$. Specifically, statistics such as the *rank correlation between the rankings generated by two protocols* or *the binary random variable indicating whether the best config was selected by a protocol* are a deterministic function of the partition. In our case, since when people do hyperparameter search, they usually only sample one partition, we set $n = 1$ and $B = 1000$. Thus, we are sampling 1000 different partitions (with replacement), calculating the statistic for each of the partitions, and displaying the 95% empirical confidence interval that results.

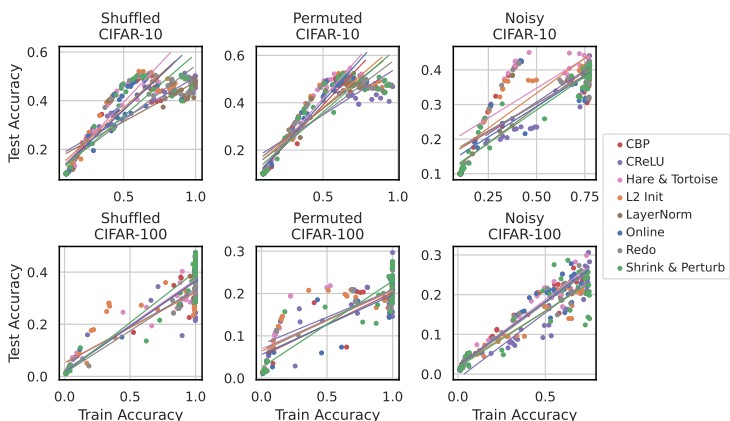

(a) Train accuracy vs Test accuracy for all configurations sampled in our study, coded by method.

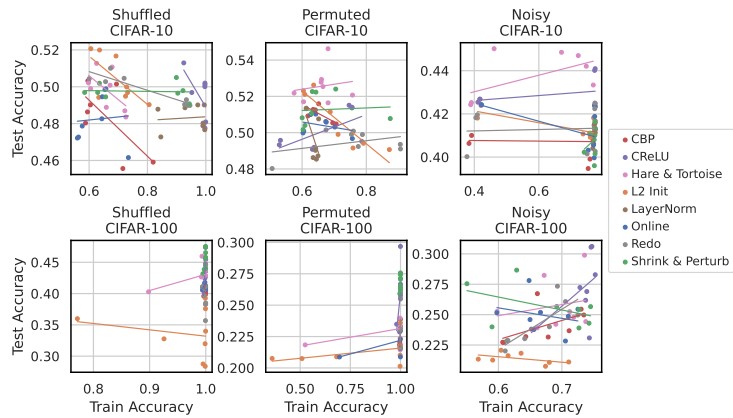

(b) Train accuracy vs Test accuracy for the top 20% of configurations sampled in our study, coded by method.

Figure 16: The relationship between Train and Test accuracy for the configurations sampled in our study, separated by method.

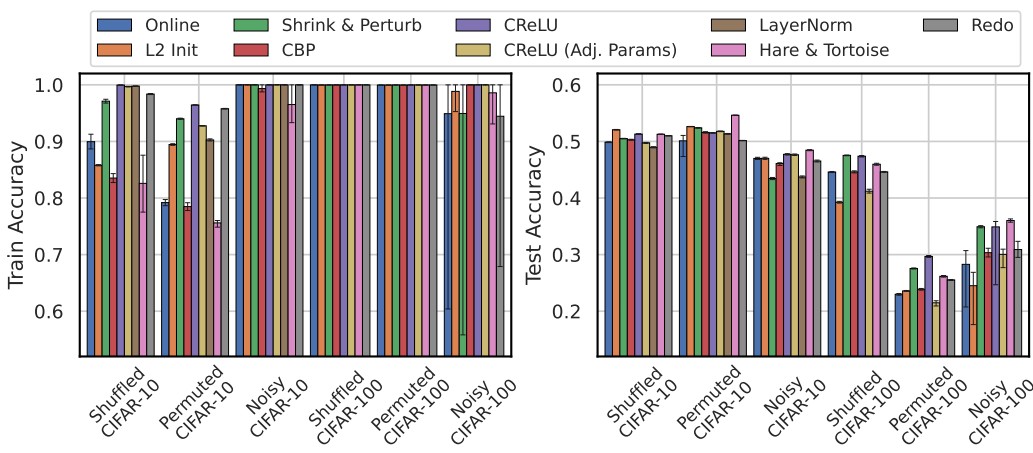

Figure 17: Results for Average Training and Testing Accuracy for each method (same as Fig. 2a) with the addition of a CReLU baseline with a smaller network such that it approximately matches the number of parameters in the network of the other methods.

