# OpenReview forum: "Experimental Design for Nonstationary Optimization"
_ICLR.cc/2025/Conference — Submitted to ICLR 2025_

### Official Review · Reviewer_MdJD · 2024-10-25

**Soundness:** 2
**Presentation:** 2
**Contribution:** 2
**Rating:** 6
**Confidence:** 3

**Summary:**

The paper conducts several experiments on selecting hyperparameters from the previous literatures for nonstationary optimization and demonstrates that using multiple streams of tasks for hyperparameters selection is the best approach among the commonly used protocols.
It also gives insight on finding configurations with good performance under low resource budget.
In addition, it shows that maintaining the training accuracy does not relate to a better generalizability in nonstationary optimization settings.

**Strengths:**

* Provides an evidence that, there is no correlation between trainability and generalizability.
* Shows large number of seeds are not necessary for finding good hyperparameters.
* Provides direction on resource-constrained experiments.

**Weaknesses:**

- The paper is not well written
    - Line 142: Duplicate of 'the added'. Same line, duplicate of 'be' in 'Should be actually be used'
    - Line 153, there is no obvious connection between this and the next sentence.
    - Line 283: 'used for selection' which I think you are referring to 'used for evaluation' instead?
    - Line 375: 'HOW MANY SEEDS DO YOU TO EVALUATE A METHOD',  miss a 'need'?
    - Line 722:  Missing the batch size for resnet-18.
    - Figure 2 is hard to interpret. One possible improvement is to add different line shapes.
    - Figure 3 does not have the legend.
    - Figure 4's line is hard to differentiate the methods. Same as figure 2, maybe add different line shape.
- The protocol 3 is considered to be critical to do lifelong learning. But there is no comparison between this protocol and the other protocols to prove that statement.

**Questions:**

Why is the number of seeds and tasks different in different type of networks, same for the gradient steps?

---

> ### Author Response · Authors · 2024-11-20
>
> We thank the reviewer for their comments. We are glad that the reviewer appreciates the insights our work provides on trainability vs generalizability and designing resource constrained experiments. We address the reviewers' questions and concerns below.
>
> **[Various typos]**
> Thank you for pointing all of these out. We have fixed them and worked to improve the writing quality of the paper.
>
> **[Protocol 3 comparison]**
> Thank you for your suggestion. We have added a comparison of Protocol 3 to the other protocols to Figure 3. We see that the method rankings generated by Protocol 3 are either worse or within the confidence interval of Protocol 2. Furthermore, even if Protocol 3 manages to generate an accurate method ranking, in section 4.5, we see that it is not good identifying the best configuration for each method. Doing well in a setting like Protocol 3 is necessary to create “lifelong” agents that have unbounded lifetimes. While we do not claim that our version of this protocol is optimal, the underperformance at least invites further research.
>
> **[Why is the number of seeds and tasks different in different types of networks, same for the gradient steps?]**
> The number of seeds and tasks are lower for the ResNet experiments just for practical compute reasons, as it takes longer to train ResNets and you can fit fewer runs onto a single GPU. The number of gradient steps per task were set based on approximately the amount of time it took for training to converge.
>
> We thank the reviewer for their valuable comments and helping improve our work. If we have addressed your questions and concerns, we would appreciate it if you could further support our work by increasing your score. If there are more questions/concerns, please let us know.

---

> > ### Comment · Reviewer_MdJD · 2024-11-22
> >
> > Thank you for updating the figures and fixing the grammar. The overall writing looks better.
> >
> > Additional Questions:
> > * `Doing well in a setting like Protocol 3 is necessary to create “lifelong” agents that have unbounded lifetimes` \
> > I agree with the reviewer EYTk, the current experiments are insufficient to claim that the protocol 3 is **necessary** to create a true lifelong agent. More experiments need to be added to investigate the properties of the a lifelong agent and how the protocol 3 can fulfill those properties. If the author can address this question, I can raise my score.
> > * For the training accuracy of the permuted cifar-100 in Fig 2. In the revision, it seems like all of the methods reached 100% accuracy where the original version is not?
> > * Same Fig 2. The legend should not explain why but what's the figure.

---

> > > ### Author Response · Authors · 2024-11-25
> > >
> > > Thank you for your response. We are glad you appreciate the improvements in the quality of the paper. We address your concerns below:
> > > **[Comment on Protocol 3]**
> > > Our point about protocol 3 being necessary to create lifelong agents that have unbounded lifetimes was perhaps more opinion than fact. We meant to convey the fact that one goal of lifelong learning is to create agents with potentially unbounded lifetimes, which necessarily means that we select hyperparameters (or meta-hyperparameters assuming a system where we adapt hyperparameters in the agent’s lifetime) using a smaller timeframe/fewer tasks than we want the agent to run. Protocol 3 is one way of doing that (not necessarily the only way, but at least one that has been proposed in the literature), and it does not do a great job at selecting the most successful agents. We simply wanted to highlight this as an invitation to the community to either create methods that work well with Protocol 3 or create new protocols that have the property that they can select successful agents that do well on lifetimes longer than what the protocol sees in the selection stage. We have updated our discussion of Protocol 3 in the paper. Thank you for your question.
> > >
> > > **[Training accuracy of Permuted CIFAR-100]**
> > > The original figure displayed a ranking for the methods rather than the actual values. For Permuted CIFAR-100, when optimizing for best average test accuracy, each method gets very close to 100% (with averages ranging from .99956 to 0.99977), essentially perfect training accuracy. Thus, the mean train accuracy is different between the methods, allowing us to create a ranking (the original Figure 2 plot), but the difference is negligible, which is why we replaced the original plot.
> > >
> > > **[Figure 2 caption]**
> > > We have updated the caption.
> > >
> > > Please let us know if this addresses your questions. If so, we would appreciate it if you could further support our work by increasing your score. If there are more questions/concerns, please let us know.

---

> > > > ### Comment · Reviewer_MdJD · 2024-11-26
> > > >
> > > > Thanks for the clarification.
> > > > Given that, I think the last sentence in the line 431 should be updated.
> > > > Will update the score.

---

### Official Review · Reviewer_EYTk · 2024-11-04

**Soundness:** 2
**Presentation:** 3
**Contribution:** 2
**Rating:** 5
**Confidence:** 3

**Summary:**

The paper presents an empirical study comparing various existing methods to mitigate plasticity loss in continual learning. The paper makes two contributions: (1) a comparison of existing methods under a unified setting, and (2) an evaluation of and suggestions for hyperparameter selection protocols, number of seeds, and train vs. test accuracy evaluation.

**Strengths:**

Continual learning is a rapidly growing and important research area. An independent comparison of existing algorithms under a unified setting aids in identifying the most effective techniques, thereby guiding future research.

**Weaknesses:**

The study setting has significant limitations and is far from realistic for continual learning. My main concern is the relevance of the proposed recipes to more realistic continual learning scenarios. Below, I summarize different ways in which the study’s setting is limited.

- Distribution shifts: The paper uses simple forms of distribution shifts, namely pixel permutation, label permutation, and noisy labels. Although these methods are frequently used in existing literature, they are highly unrealistic. Pixel permutation, for example, never occurs in real-world scenarios (and the use of CNNs for these types of images is questionable). Moreover, these distribution shifts are simple to address, as they can be resolved by adjusting weights in the first or last layer. While benchmarking is an unresolved problem for continual learning and good benchmarks are still limited in the literature, more realistic distribution shifts have been proposed in other works (e.g., see the first three environments studied in Dohre et al., 2024), which could be used in the current paper.

- Test/Train sets: In a realistic continual learning setting, there is no separate test or train set. Instead, there is a single, continuous stream of incoming data, to which a model adapts. Cumulative online loss serves as the primary performance measure.

- Random seeds: True continual learning settings do not involve random seeds for the same reason discussed above, especially in scenarios with sufficiently long data streams.

- Hyperparameter selection:  Continual learning is best achieved through continual optimization, which includes algorithms for continual hyperparameter optimization. Here, hyperparameters (e.g., learning rate) are optimized and updated at every training step, over the whole lifetime of agent. See, for example, IDBD, Hypergradient Descent, and MetaOptimize.

I understand that these limitations are also present in many existing works. While evaluations in limited settings are acceptable in experimental sections for papers introducing new algorithms, such limitations are insufficient for an empirical study aiming to provide guidelines for future research.

Lastly, the scale of the experiments and models used is relatively small for a fully empirical study.

**Questions:**

Could the authors conduct experiments on more realistic benchmarks from the literature or propose a new, more realistic benchmark?

In Fig. 9, the Online baseline (with no LoP technique) outperforms some LoP algorithms. What is the reason for this? Could it be due to insufficient hyperparameter tuning?

The paper suggests that test accuracy rankings differ significantly from train accuracy rankings. Could the authors quantify this gap for different algorithms? Although the rankings change, the actual performance difference might be minor.

How would CReLU perform compared to other methods if it used the same number of weights?

What do the two curves in Fig. 3 represent? There are no legends.

There are also a few typos on page two.

---

> ### Author Response · Authors · 2024-11-20
> **Response 1/2**
>
> We thank the reviewer for their comments. We are glad that the reviewer appreciates that our work provides a unified and independent evaluation of the works done by the plasticity community. We address the reviewers' questions and concerns below.
>
> **[Use of unrealistic benchmarks]**
> Our use of the datasets and nonstationaries studied was motivated by the fact that these are the types of settings that are in fact used in plasticity research, and we wanted to give empirical recommendations to the plasticity community. Several recent works [1,2,3,4,5] have used these settings as a way to simplify and study the core problem, including Dohare et al., 2024. We also do acknowledge the need for newer and more difficult settings in our discussion of our results in Section 4.3. We also would like to point out that the Noisy distribution shift is representative of a fairly realistic distribution shift where you get higher quality data over time, and is encountered in both the supervised setting and the reinforcement learning setting. Given that these are commonly used settings and we do include a realistic setting, we do not think this should be a reason to reject the paper.
>
> **[Use of Train/Test sets]**
> Our work aims to facilitate the research that people have started doing in nonstationary optimization/plasticity. While the setting with only a stream of data is a valid continual learning setting, there are several use cases, both in research and in real systems, where there would be a train/test split in order to quantify the generalization capabilities of the system. Many works [1,2,3.4,5] in both the general continual learning field and in the plasticity field have also introduced such a split, as it allows us to better probe the abilities of these systems.
>
> **[Use of random seeds]**
> Similar to our point above, we agree that when continual learning systems are deployed, it probably would not make sense to have random seeds. The goal of our work, however, is to facilitate the research on these systems, and being in order to evaluate whether the methods we are developing are overfitting to a single task sequence or initial random conditions, we need to evaluate with multiple seeds, as has been done throughout the continual learning literature [1,2,3.4,5].
>
> **[Continuous hyperparameter selection]**
> Continuous hyperparameter selection is a separate research problem within continual learning. It would most likely improve the performance of the methods we study, but as we are trying to facilitate and provide recommendations for research in plasticity, applying those methods in our work when they are not used in the plasticity community would make our recommendations less applicable and helpful to this community.
>
> **[Limited settings]**
> Our goal with this work was not to create a realistic continual learning system or to provide empirical guidelines on how to create a realistic system. It was to provide empirical research recommendations to the growing plasticity community. The work in this community tries to create methods that can deal with nonstationary optimization and to study it, we need to make simplifying assumptions and make sure we are providing repeatable results. This could lead to limited settings, but is also useful to make progress on the problem.
>
> **[Small scale of experiments]**
> As a part of this study, we ran 27,600 trials across multiple datasets, distribution shifts, architectures, and methods. The ResNet architecture is also one of the larger architectures studied in plasticity research. We believe that this is an acceptable scale for such an empirical study.
>
> **[Online algorithm doing better than baselines]**
> We are not sure of how well the online baseline was tuned in prior work, but one of the contributions of our work is showing that a well tuned online baseline (with L2 regularization, we allow L2 regularization for all methods except L2-Init) can be quite competitive and beat out several existing plasticity methods.
>
> **[Gap between train and test accuracies for different methods]**
> We have moved the original Figure 2 to the appendix (it is now Figure 14), and replaced it with a bar chart (Figure 2a) showing the train and test accuracies achieved by each method. Here, we can see that it is not just the ranking but the actual performance that can change significantly going from train to test. For example, methods such as L2-init and Hare and Tortoise are worse than the best methods on training accuracy by 10-20% on both Shuffled CIFAR-10 and Permuted CIFAR-10, and yet they are the best and second best methods for test accuracy on those two settings. We can also see in Figure 2b that several of the method rankings generated when selecting for the best train accuracy are actually anti correlated with method rankings generated when selecting for test accuracy.

---

> > ### Author Response · Authors · 2024-11-20
> > **Response 2/2**
> >
> > **[CReLU with equal number of weights]**
> > We are currently running this experiment and will update this post and the manuscript when we have the results. Likely for MLP, the results would not be significantly different, as to get an equal number of parameters, you can reduce the hidden size from 128 to 120. For ResNet, the results might change, as you need to reduce the number of filters from 64 to 45.
> >
> > **[Figure 3 Legend]**
> > We have updated Figure 3 to be a bar plot and added the missing legend. We also added Protocol 3 to the evaluation. Each bar represents the performance of one of the protocols described in Section 3.3. Thank you for pointing out the missing legend.
> >
> > **[Typos on page two]**
> > Thank you for pointing them out, we have updated the manuscript to fix the writing errors.
> >
> >
> > We thank the reviewer for their valuable comments and helping improve our work. If we have addressed your questions and concerns, we would appreciate it if you could further support our work by increasing your score. If there are more questions/concerns, please let us know.
> >
> > [1] Dohare, Shibhansh, et al. "Loss of plasticity in deep continual learning." Nature 632.8026 (2024): 768-774.
> > [2] Kumar, Saurabh, et al. "Maintaining Plasticity in Continual Learning via Regenerative Regularization." Conference on Lifelong Learning Agents. PMLR, 2024.
> > [3] Lee, Hojoon, et al. “Plastic: Improving Input and Label Plasticity for Sample Efficient Reinforcement Learning.” Advances in Neural Information Processing Systems, vol. 36, 2024.
> > [4] Elsayed, Mohamed, et al. "Weight Clipping for Deep Continual and Reinforcement Learning." Reinforcement Learning Conference.
> > [5] Lee, Hojoon, et al. "Slow and Steady Wins the Race: Maintaining Plasticity with Hare and Tortoise Networks." Forty-first International Conference on Machine Learning.

---

> ### Comment · Reviewer_EYTk · 2024-11-22
> **After reading the authors’ response:**
>
> I appreciate authors’ responses and their modifications, which I think has improved the paper’s presentation. Specifically, the revision has addressed most questions, except for the most important question above (i.e., my first question).
>
> On the negative side, I think the revision (and the response) falls short of concrete improvements towards addressing the weaknesses mentioned above.  In particular, the first contribution of the paper, as mentioned is the abstract, is the evaluation and comparison of existing algorithms, which I think can be potentially valuable; however the authors performed this comparisons on some of the simplest and most unrealistic benchmarks available in the literature. This significantly weakens the intended contribution (given that this is a fully empirical paper). The argument that these benchmarks are also used in existing works is not convincing, because those works also include several better and more realistic benchmarks that could have been used here (refer to the weaknesses above for pointers and detailed discussion).
>
> The fact that the online baseline in the reported experiments perform as well as an average LoP method, potentially suggest that **no** Loss of Plasticity occurs in the tested environments, rendering LoP studies irrelevant on the tested benchmarks (otherwise the online-baseline’s relative performance would have considerably decayed over time due the *lost plasticity*, by definition of LoP).
>
> As another weakness regarding contribution, the last line of the abstract claims to “provide concrete recommendations and analysis that can be used to guide future research”. While the paper provides some interesting insights that can be useful for future research, I think many of these recommendations are not actually relevant for realistic continual settings for the reasons detailed in the weakness section above; and overall they do not direct the field towards addressing challenges that really matter in long term.
>
> On the plus side, some of the insights and recommendations are interesting, especially for the simplified setting of non-stationary supervised learning. The gap between train and test can be important (I have personally observed it in some of my older continual learning experiments). I  also think sensitivity analyses like Fig. 8 are useful for future use of these methods.
>
> Regarding my score, the grounds for rejection are not due to mistakes or inaccuracies in the paper but rather my judgment of the overall contribution of the paper, considering the rather high bar for ICLR. This judgment is, of course, influenced by my subjective perceptions of what matters in LoP research, which may prove inaccurate with the test of time. I actually lean toward a score of 4 (but unfortunately the possible scores are quantized at 3 and 5). Given all the improvements in the revision and the remaining concerns, I am keeping the score of 3 and reducing my confidence to 2, favouring judgment about the contribution from other expert reviewers.

---

> > ### Author Response · Authors · 2024-11-25
> >
> > Thank you for your response. We are glad you appreciate the improvements in the quality of the paper and that you feel that we have addressed most of your concerns. We also would like to let you know that we have finished running preliminary searches with CReLU with the network size adjusted to maintain the same number of parameters. For time constraints, we ran a search with 30 configurations instead of 40 configurations. We found that the MLP results are approximately the same, with a slight decrease in performance in Permuted CIFAR-10 training accuracy and Shuffled CIFAR-10 test accuracy. For ResNet, the training performance is matched by the smaller network, but the test performance is significantly lower across all settings. You can see the full results in Appendix D.6.
> >
> > We address your remaining concerns below:
> >
> > **[Unrealistic benchmarks]**
> > We have launched experiments for Continual Imagenet to extend our study, however, we would like to point out that the benchmarks used in our study are very similar to the types of benchmarks that are used in the literature. The reviewer mentions Dohare et al, 2024 as a source of realistic benchmarks. The three continual supervised learning benchmarks used in Dohare et al, 2024 are (1) Continual Imagenet: in this benchmark, the network faces a series of binary classification tasks between two Imagenet classes. The images are scaled down from the typical Imagenet size to 32x32 pixels, and after every task, the last layer is reset. We argue that our settings are at the very least as challenging as this setting, given our tasks have more classes and we don’t do partial resets between tasks. (2) Class incremental CIFAR-100: In this setting, the network gets access to data from more and more classes over time. While it is a different type of distribution shift, we’d argue that our CIFAR-100 experiments are at a similar level of difficulty. The noisy CIFAR-100 experiment gives access to different parts of the dataset over time (not accumulating the data like class incremental CIFAR-100), and changes the noisiness in the label space over time. (3) Permuted Input MNIST: This is simply a less challenging version of our Permuted CIFAR-10 and Permuted CIFAR-100 experiments.
> >
> > While we appreciate the need for more realistic benchmarks, we believe that the settings included in our study are at a comparable difficulty level to what has been used in the literature for plasticity research, both in Dohare et al, but also several of the other citations provided above. Our goal with this paper is not necessarily to propose realistic continual learning settings, but to provide recommendations to researchers working in plasticity research, which does not necessarily happen in realistic continual learning settings.
> >
> > **[Performance of Online baseline]**
> > We want to emphasize that our online baseline (as well as all the methods other than L2-Init) also has L2-regularization added, as a basic defense against loss of plasticity. Our reasoning for this was that (1) L2-regularizatiton is a ubiquitous technique in deep learning at this point (2) previous works have shown that an increase in weight norm essentially always leads to loss of plasticity. Given the simple and ubiquitous nature of L2 regularization, we made some form of L2 regularization default for all methods. This change makes a well tuned online baseline significantly stronger, which we believe is still a useful thing to point out to the community about the methods being proposed. If they cannot outperform simple L2 regularization, that should be noted.
> >
> > We hope that our responses and additional experiments have addressed your concerns. If so, we’d greatly appreciate it if you would increase your score.

---

> > > ### Author Response · Authors · 2024-12-02
> > >
> > > Thank you for your response. We would like to let you know that we have finished running experiments with Vision Transformers on the Incremental CIFAR-100 setting described in [1]. The main results are summarized in Global Response #2, and they are all generally in line with our previous conclusions. We thank the reviewer for pushing us to do these experiments as they did make the evidence for our conclusions stronger. We now have experiments that provide coverage on two realistic data shifts (Noisy and Incremental CIFAR-100) and two realistic architectures (ResNets and Vision Transformers). If we have addressed your concerns about our work, we ask that you please support our work further by raising your score.
> > >
> > > [1] Dohare, Shibhansh, et al. "Loss of plasticity in deep continual learning." Nature 632.8026 (2024): 768-774.

---

> > > > ### Author Response · Authors · 2024-12-02
> > > >
> > > > Dear Reviewer,
> > > >
> > > > Since this is the last day where reviewers are allowed to communicate with the authors, we kindly ask you to let us know whether our additional results and changes have addressed your concerns, and if so, to please adjust your score accordingly.
> > > >
> > > > Thank you,
> > > > The Authors

---

> > > > ### Comment · Reviewer_EYTk · 2024-12-03
> > > >
> > > > Thanks to authors for their effort in addressing the raised concerns about realistic benchmarks, and for conducting experiments on Continual CIFAR100. Given these improvements, I raise my score from 3 to 5.

---

### Official Review · Reviewer_2oWR · 2024-11-04

**Soundness:** 3
**Presentation:** 2
**Contribution:** 3
**Rating:** 6
**Confidence:** 4

**Summary:**

The authors investigate the experimental procedures that are used to evaluate algorithms in continual learning settings.
The submission points out that these practices are usually unaddressed or only implicitly addressed in current experimental work in continual learning.
By focusing on a well-curated set of datasets, nonstationarities, methods and architectures, the submission poses a critical analyses of these practices and implicit assumptions.
A few interesting conclusions include that maintaining trainability may not be indiciative of generalizability, and that several tasks may be needed to evaluate a set of hyperparameters for continual learning.

I am currently rating this paper as marginally below acceptance. However, I am willing to increase my score if some of the concerns below are addressed.

**Strengths:**

- The problem being addressed in this submission is timely, understudied and important.
- Empirically, the submission poses and answers several important questions regarding best practice for non-stationary optimization.

**Weaknesses:**

- While most of the paper provides concrete takeaways which question current practice, there are some inconclusive results that could use further clarification
- Several of the results aggregate performance across several different categories of methods (e.g., architectural(crelu), regularization and resetting). A few analyses on individual methods, non-aggregated, would improve the empirical results and add clarity.
- The presentation of the paper is mostly good, but with room for improvement (see below).

**Questions:**

- Figure 2 (presentation): It is difficult to infer conclusions from this plot due to the number of baselines and settings being compared in a single graph. I wonder if this information is not better represented in a table, because the relative ordering of the settings (on the x-axis) and the placement of MLP next to ResNet does not have any semantic meaning in the presentation of these results.

- Figure 2 (results): I am surprised that there is so much variation in the results for different combinations of methods and nonstationarities. One thing potential problem is that the ranking is too sensitive to statistically insignificant performance differences. Do you know how these results would look like if average test accuracy was reported instead?

- Figure 3: The lines in these graphs are not labelled, I assume one is for protocol 1 and the other is for protocol 2? If that is the case, I do not see that large of a difference between the two (except on shuffled CIFAR-10). Thus, I am uncertain about the conclusion that "protocol 2 transfers better to the unseen data". The conclusion suggested at the end of Section 4.1 does not seem well-supported by this data.

- Clarification for statistical bootstrapping: what exactly is being resampled for the estimate? It is not clear how "resampling the seeds" means, because bootstrapping usually involves resampling from some data to construct an estimator.

- Clarifying seeds in Section 4.1: Why are the total number of seeds quoted (n=20) unequal between protocol 1 and 2? It seems like that the seeds are partitioned between model selection and evaluation? As I understand the second paragraph, 10 seeds are used for model selection and 10 seeds are used for evaluation, yielding the total of 20? But in that case, why does protocol 2 only use 5 seeds?

- There seems to be no clear takeaway in Section 4.2: it would be helfpul to also investigate the contributing factors for generally well-performing methods. For example, are the methods performant because they are more robust to hyperparameters (and hence, protocol 2 can easily identify good hyperparameters)? I do not think Appendix C answers this question.

- Section 4.3: The strong conclusion here is valuable. I wonder if this conclusion depends on the plasticity-preserving method. I am not able to tell from Figure 4, but presumably some methods may better correlate train and test accuracy, which would be hidden in this combined analysis.

- Section 4.4: Again, I wonder if the number of seeds needed to identify a good hyperparameter configuration depends more strongly on the method used for training, rather than the aggregate analysis.

** Minor Comments
- Many instances of "boostrapping" should be replaced with "bootstrapping"

---

> ### Author Response · Authors · 2024-11-20
> **Response 1/2**
>
> We thank the reviewer for their comments. We are glad that the reviewer finds our work timely, understudied and important. We address the reviewers' questions and concerns below.
>
> **[Inconclusive Results]**
> We have updated the paper with more concrete takeaways (specifically for sections 4.1, 4.2, and 4.3). If there are other parts that the reviewer would like expanded on, please let us know.
>
> **[Non-aggregated analysis]**
> We thank the reviewer for their useful suggestion. We have added Figure 15 and Table 3 that do a non-aggregated analysis of the results in Section 4.3 and Figures 10 and 11 that show a non-aggregated analysis of the results in Section 4.4 (please see below for further discussion). Furthermore, Figure 12 (present in the original manuscript) shows a non-aggregated analysis of the results in 4.6.
>
> **[Improvement of Figure 2]**
> We have moved this Figure to the appendix (Figure 14), and replaced it with a plot showing the actual values of the train and test accuracies used to generate the rankings, and a plot showing how much the different rankings correlate with each other. From this Figure, we can see that while there are a few overlaps in the error bars (representing 95% empirical confidence intervals), most of the results would stay fairly stable.
>
> **[Figure 3 results]**
> We have updated Figure 3 to be a bar plot and added the missing legend. We also added Protocol 3 to the evaluation. Each bar represents the performance of one of the protocols described in Section 3.3. You are right that perhaps our original claim was too strong. Protocol 2 is not outright better than the other protocols, but it is the only one that is not outright beaten by another protocol (taking into account the confidence intervals), and it is the outright best on Shuffled CIFAR-10. It outright beats Protocol 1 on 2 of the 6 settings, and approximately matches performance (within CI) on the other 4. Protocol 3 similarly is outright worse on 2 settings compared to Protocol 2, and within the CI on the other 4. Thus, we argue that there is not much advantage to be gained from using Protocols 1 or 3, and at least a few settings where it is disadvantageous to do so. We have updated Section 4.1 with this discussion.
>
>
> **[Clarification for statistical bootstrapping]**
> With statistical bootstrapping, we sample $B$ datasets of size $n$, compute some statistic using each of these datasets, and use the empirical distribution over the statistic to create confidence intervals or compute standard errors for the sample mean of the statistic. In our case, we are sampling partitions over the seeds, $P$, and using the partition to estimate some statistics that are a function of this partition, $f(P)$. Specifically, statistics such as the *rank correlation between the rankings generated by two protocols* or the *binary random variable indicating whether the best config was selected by a protocol* are a deterministic function of the partition. In our case, since when people do hyperparameter search, they usually only sample one partition, we set $n=1$ and $B=1000$. Thus, we are sampling $1000$ different partitions (with replacement), calculating the statistic for each of the partitions, and displaying the 95\% empirical confidence interval that results. We have added a discussion of this to the appendix.
>
> **[Clarifying seeds in Section 4.1]**
> Of the 20 seeds, 10 are used to create a held out ranking. These are not available to any of the protocols when doing model selection or evaluation. The idea is that they represent unseen task sequences, and we can see how the method rankings generated by each protocol transfer to unseen task sequences. Protocol 1 uses the validation accuracy of all 10 of the available seeds to select the best configuration for each method and uses the test accuracy of those same seeds to rank/evaluate them. Protocol 2 uses the test accuracy of 5 of the 10 available seeds to select, and the test accuracy of the other 5 to rank/evaluate. We also add Protocol 3 to our evaluation, which uses the test accuracy of all 10 seeds over the first 10% of tasks to select, and the test accuracy of all 10 seeds over the latter 90% of tasks to rank/evaluate. We have updated the writing in 4.1 to make this procedure more clear.

---

> > ### Author Response · Authors · 2024-11-20
> > **Response 2/2**
> >
> > **[Takeaways from 4.2]**
> > We have updated the writing in 4.2 and updated Figure 2 to provide more clear takeaways. Specifically, (1) we see that a well tuned Online baseline actually performs surprisingly well (2) Training accuracy starts to saturate on 4 out of the 6 settings we study (3) For test performance, Hare and Tortoise and CReLU do well across almost all settings, Shrink and Perturb does well on with ResNet, and L2-init does well with MLP and badly with ResNet.
> > Since we are selecting the best configuration for each method, hyperparameter sensitivity probably does not have a huge effect on these results. Further analysis of why each method works to the level it does, while useful, we believe is out of scope for this paper.
> >
> > **[Non aggregated analysis of Section 4.3]**
> > We have added Figure 15 and Table 3 that present a non aggregated version of the results in Section 4.3. Similarly to the full version, when considering every configuration, there is a positive correlation. When restricting to the top 20% of configurations, the correlation disappears for most of the methods/settings. Thank you for this suggestion.
> >
> > **[Non aggregated analysis of Section 4.4]**
> > We have added Figures 10 and 11 to Appendix D.1 that present a non-aggregated version of the Figures in Section 4.4. They show that there is in fact some difference between the methods in terms of how sensitive they are to the number of seeds used. For example, other than the Online baseline on Noisy CIFAR-100 with ResNet, the configuration evaluated as best in the search is very likely to at least be a top 3 configuration even with 1 seed (Figure 10b). We can also see some differences in how many seeds it takes to accurately rank the different configurations for a method in Figures 11a and 11b. Thank you for this suggestion.
> >
> > **[Bootstrapping typo]**
> > Thank you for pointing it out, it has been fixed.
> >
> > We thank the reviewer for their valuable comments and helping improve our work. If we have addressed your questions and concerns, we would appreciate it if you could further support our work by increasing your score. If there are more questions/concerns, please let us know.

---

> > > ### Author Response · Authors · 2024-11-25
> > >
> > > Hello, as the discussion period is ending soon, we would like to hear your thoughts about our updates and our response to your review. If we have addressed your concerns, we would appreciate it if you could further support our work by increasing your score.

---

> > > > ### Comment · Reviewer_2oWR · 2024-11-26
> > > >
> > > > I appreciate the authors' detailed reply. The clarity on some points has definitely improved, some comments:
> > > >
> > > > - Improved takeaways:
> > > > Unfortunately, I cannot tell what has changed in the submission. You should highlight changes made in the updated submission with blue text. It is almost impossible to tell what has changed, apart from the table. As a result, I am unable to truly appreciate the changes the authors made.
> > > >
> > > > - Non-aggregated results: while I see the new figures in the appendix, I am not sure what to take away from them. I would expect that certain methods that are more robust or performant would have higher probability of finding a successful configuration. Unfortunately, this not currently obvious.
> > > >
> > > > Reading the other reviews, I would agree that the presentation of the paper is still a weakness (although it is improved). My suggestion (for a potential camera-ready/revision/resubmission) is to narrow the focus of the questions in the main text. The introduction lists 6 questions, the discussion at the end lists 8 takeaways. I would limit both of these to exactly three of whichever you think are the most interesting. One example of an interesting takeaway is that train does not correlate with test for loss of plasticity. In contrast, referencing appendix results in the takeaways is uninteresting and  out of place. Limiting the main questions being asked will improve presentation, and allow the conclusive results to shine without being held-back by other less conclusive results (which can still populate the appendix).
> > > >
> > > > Overall, I think the authors have improved the paper. In particular, I would disagree with the other reviewer: the choice of datasets/nonstationarities is fine. Not only is it a common point of reference with respect to published literature, I also think it is sufficiently complex for benchmarking (although, clearly not realistic). I am going to continue to monitor discussion with other reviewers. It is possible that I will marginally increase my score, but I will see after the discussion period.

---

> > > > > ### Author Response · Authors · 2024-11-28
> > > > >
> > > > > Thank you for your response. We are glad you appreciate the improvements in the clarity of the paper. We have also updated the paper and highlighted major text changes from the original submission and new figures in blue to more easily spot the differences.
> > > > >
> > > > > **[Presentation of takeaways]**
> > > > > We have reorganized our question list in the Introduction and our takeaways list in the Discussion. Specifically, we group the discussion around the number of seeds, tasks, and configurations into a single discussion of how to design resource efficient hyperparameter searches. The discussions around the different hyperparameter search protocols and training vs testing plasticity remain as their own paragraphs. Thank you for the suggestion on how to give our conclusions a better focus, and please let us know if you have more comments.
> > > > >
> > > > > **[Non aggregated results takeaways]**
> > > > > Unfortunately, the nature of the non aggregated results is that there are quite a few methods being tested on several settings, and thus there is a lot of information to showcase which is difficult to fully describe in text. We think that some of the most concrete takeaways from the analyses are (1) from Figure 10b, for essentially all methods on all settings except for Online on Noisy CIFAR-100, the hyperparameter configuration selected will be a top-3 configuration even when using 1 or 2 seeds (2) from Figure 10c, the best configuration will be evaluated as a top 3 configuration in the search with very few seeds for every case except Noisy CIFAR-100 and Online on Permuted CIFAR-10 (3) Figures 11a and 11b show approximately the ordering of how sensitive the configuration rankings are for different methods to the number of seeds used in model selection. Although there is a lot of overlap, you can still see some separation between methods (4) Figure 11c shows the approximate range expected for the reported test accuracy of a method given a certain number of seeds used for selection. For most methods on most settings, there isn’t a very large range even with a small number of seeds, but you can see a few methods with large ranges for small numbers of seeds that need more seeds to narrow the range (5) Figure 12 shows that there isn’t much difference between methods for the expected improvement per extra configuration or range of expected final test accuracy (6) Figure 16b and Table 3 show that for most methods across all settings, when focusing on the top configurations for each method, there isn’t a statistically significant positive correlation between train and test accuracy. In fact, the only place where such a relationship exists is for ReDo with ResNets on Noisy CIFAR-100 (the other statistically significant positive correlations in the table are essentially vertical lines that are not well defined correlations). We have updated the manuscript and highlighted the text with these discussions.
> > > > >
> > > > > We again thank the reviewer for their valuable comments and helping improve our work. If there are more questions/concerns, please let us know. If we have addressed your questions and concerns, we would appreciate it if you could further support our work by increasing your score.

---

> > > > > > ### Author Response · Authors · 2024-12-02
> > > > > >
> > > > > > Hello, we would like to inform you of our expanded results using Vision Transformers on Incremental CIFAR-100. The main results can be seen in Global Response #2. We would greatly appreciate it if the reviewer can take a look at these results as well as our new manuscript which we reorganized for clarity and let us know if they have any further questions or concerns. If not, we would appreciate it if you could further support our work by increasing your score.

---

> > > > > > > ### Comment · Reviewer_2oWR · 2024-12-02
> > > > > > >
> > > > > > > I appreciate the changes to the manuscript. In particular, the increased focus on the three key takeaways in the last section helps highlight the contribution made by the submission. I will update my score accordingly (5->6) and support acceptance.

---

### Official Review · Reviewer_wsyP · 2024-11-10

**Soundness:** 2
**Presentation:** 2
**Contribution:** 3
**Rating:** 5
**Confidence:** 5

**Summary:**

In this paper, the authors study the experimental design in non-stationary optimization. They explore various aspects of experiment design, from hyper-parameter selection protocols to the number of seeds needed for hyper-parameter selection. The paper contains some interesting results about hyper-parameter selection, number of seeds, experiment protocols, etc.

**Strengths:**

This work is novel because there has not been any explicit focus on empirical design in plasticity research. It could be an important contribution to the community as it could speed up progress and provide a more unified focus for the field.

The studies about hyper-parameter selection protocol are useful as they could help develop methods that overfit less.

Results show that the community needs to focus on test plasticity, which is interesting and needs to be evaluated further.

The study about the number of seeds needed for hyper-parameter selection and evaluation of a method is good, and it could save computation on many future experiments.

**Weaknesses:**

The paper is not well-written and contains a lot of small errors, which reduces my confidence in the quality of the paper and the results presented therein. For example, there are no labels in Figure 3, and line 424 says "Figure 5b and 5b", a typo in the title of section 4.4, among many others. I suggest the authors spend some more time on writing (maybe by using Grammarly) and making sure the presentation is up to the mark.

The experiments with ResNet and permuted CIFAR are not useful. When inputs are permuted, all the spatial information in the image is lost. In such a case, convolutional networks like ResNet-18 are not useful. These experiments have not been done in the community either, people have only used feed-forward networks with Permuted Input experiments. The ResNet experiments on Permuted CIFAR-100 should be removed from the paper.

What is plotted in Figure 2? Is it the performance of the best hyper-parameter configuration or the average performance across hyper-parameters? And what does "best" mean? Highest training accuracy or test accuracy or train for Figure 2a and test for Figure 2b?

The results presented in Figure 4 are used to argue that "improving training does not end up correlating with ... improving model performance" (lines 370-371). But that is not what the figure shows. Figure 4a clearly shows that there is a positive correlation between training accuracy and test accuracy. What Figure 4b shows is that for the best hyper-parameters, there is a weak or no correlation between the two. That just means that after a point, trainability does not improve generalizability. But that does not mean "improving training does not ... correlating ... improving model performance". The claim on lines 370-371 and the 3rd bullet point in section 5 need to be changed.

In the introduction and Section 4.5, the paper asks, "How many tasks do you need to include in your training sequences?" The answer should be infinity because we are in a lifelong learning setting. Do the authors mean, "How many tasks do you need to include in your training sequences **for tuning hyper-parameters**?" or does that statement mean something else?


This paper can be a good contribution to the community, but it is not up to the mark yet. I'm willing to increase my score if the authors address my concerns.

**Questions:**

What are the labels for both lines in Figure 3?

What happens if you plot training loss and test loss in Figure 4? I suspect that could reveal more correlation. Particularly for ResNet in Figure 4b, as it gets to 100% train accuracy in most cases.

In Figure 5d, do you mean p(Method ranking **does not** change) instead of p(Method ranking changes)?

---

> ### Author Response · Authors · 2024-11-20
>
> We thank the reviewer for their comments. We are glad that the reviewer finds our work novel, timely, and useful for the plasticity community both in terms of what we should focus on and in terms of helping make efficient experiments. We address the reviewers’ questions and concerns below.
>
> **[Fixing the writing errors]**
> We thank the reviewer for pointing out the writing errors in our manuscript. We have gone through and made the corrections.
>
> **[Permuted CIFAR experiment with ResNet]**
> While we agree that permuted CIFAR-100 with ResNet is not typically seen in the community and that it would seem ResNets would not be well suited for the task, we chose to include them because we saw the networks did seem to be able to achieve perfect training accuracy and a nontrivial test accuracy. The permuted input task is already a very artificial one, only used to probe the abilities of continual learning methods. Despite this, there is still some structure there that ResNets are able to learn. If the reviewer strongly feels that this experiment detracts from the paper, we can do so, but we would like to emphasize that doing so would not affect our claims and analysis.
>
> **[Figure 2 explanation]**
> In Figure 2, we perform Protocol 2 for model selection and evaluation. We use half the seeds to find the best hyperparameter configuration for each model, and the other half to evaluate the best configurations and create a ranking of the methods. We show the ranks of the methods for each of the settings in our paper. The left plot shows the method rankings when selecting/evaluating for best training accuracy, and the right plot shows the rankings when selecting/evaluating for best testing accuracy. In the updated manuscript, we have moved this Figure to the appendix (Figure 14), and replaced it with a plot showing the actual values of the train and test accuracies used to generate the rankings, and a plot showing how much the different rankings correlate with each other.
>
> **[Updating the claim in Figure 4 about train and test accuracy not correlating with each other]**
> The reviewer is correct that we need to be more precise about this claim. What we meant to convey is that trainability correlates with generalizability *only up to a point*, after which continuing to improve trainability does not end up correlating with the end goal of improving model performance. This suggests that (1) for the types of settings presented in this study (which are representative of what is currently studied in the literature), we should shift our focus from improving trainability to the problem of improving generalizability. (2) Studying trainability could still be a valuable problem, but we should find harder settings to do so.
>
> We have updated the discussion of this claim (in Section 4.3) in the paper and thank the reviewer for the valuable suggestion.
>
> **[Clarification on “How many tasks do you need to include in your training sequences?”]**
> Yes, we meant how many tasks do you need to include when doing hyperparameter selection. This has been updated in the manuscript.
>
> **[Legend for Figure 3]**
> We have updated Figure 3 to be a bar plot and added the missing legend. We also added Protocol 3 to the evaluation. Each bar represents the performance of one of the protocols described in Section 3.3. Thank you for pointing out the missing legend.
>
> **[Figure 4 with loss instead of accuracy]**
> We have added this Figure to the Appendix (Figure 12). While some of the trends are different, there are still only two settings that have a statistically significant positive correlation between train and test accuracy for the top configurations. Thank you for the suggestion.
>
> **[X axis label on Figure 5d]**
> Yes, sorry for the mixup. We have updated the label.
>
> We thank the reviewer for their valuable comments and helping improve our work. If we have addressed your questions and concerns, we would appreciate it if you could further support our work by increasing your score. If there are more questions/concerns, please let us know.

---

> > ### Author Response · Authors · 2024-11-25
> >
> > Hello, as the discussion period is ending soon, we would like to hear your thoughts about our updates and our response to your review. If we have addressed your concerns, we would appreciate it if you could further support our work by increasing your score.

---

> > > ### Comment · Reviewer_wsyP · 2024-11-26
> > >
> > > I appreciate the time and effort the authors have put into the rebuttal. It has improved the clarity of the paper, and it addresses most of the issues I raised.
> > >
> > > I read other reviews, and I think the most significant remaining issue for the paper is the choice of benchmarks. I realize that these benchmarks are commonly used in the literature and that the noisy label benchmark is realistic. However, the paper has two unrealistic benchmarks: permuted inputs and shuffled labels. I understand that having a simple benchmark is good for understanding the problem. But I think we as a field should start moving towards more realisitic benchmarks and networks. Given that this paper aims to give guidelines on conducting experiments on non-stationary optimization, I think it should move the field towards more realistic settings. I suggest that the authors replace the permuted input benchmark with class-incremental CIFAR-100. And if you're open to it, include some non-stationary RL environments, like the one used in Figure 3 by Dohare et al. 2024. And I'd suggest including vision transformers in the experiments. To save compute, having ResNet for the Noisy label experiment and ViT for Class-incremental experiments would be okay.
> > >
> > > Because of this issue, I'd not recommend this paper for acceptance at this moment. However, I strongly encourage the authors to change the benchmarks and networks and submit the work to the next venue. Perhaps the authors should consider submitting this work to a Journal as this type of work requires a lot of back and forth with the reviewers and large changes in the paper, which are not possible in the timeline of conferences but possible in a Journal.

---

> > > > ### Author Response · Authors · 2024-11-28
> > > >
> > > > Thank you for your response. We are glad you appreciate the improvements in the quality of the paper and that you feel that we have addressed most of your concerns.
> > > >
> > > > Regarding the use of “realistic” settings, we have launched vision transformer experiments for class incremental CIFAR-100, and hope to have them finished before the end of the discussion period (although not before the end of the manuscript update deadline). We would like to point out that, as you have noted, we do have a “realistic” shift in the Noisy CIFAR-100 and Noisy CIFAR-10 experiments. We’d also put forth two other points: (1) the other settings being mentioned even in Dohare et al. can also be argued as not realistic. For the supervised learning experiments, aspects such as scaling down larger images to 32x32, using simple tasks such as binary classification or tasks with data on the order of ~1200 samples, resetting the head of the network on each task or doing early stopping on tasks all can be argued as being unrealistic continual learning scenarios. Despite this, the experiments in the paper are still very useful for learning about plasticity. (2) There are always more benchmarks that can be added to a paper such as ours, and thus it can always be a reason for rejection. We believe, however, that our work contains enough interesting messages for the community and is thus worthy of publication.

---

> > > > > ### Comment · Reviewer_wsyP · 2024-11-29
> > > > >
> > > > > I agree that it is unclear where exactly we should draw the line for "realistic" experiments. But I think we can all agree that permuting pixels is significantly more unrealistic than downsampling images. And, as I said before, I understand very well why simple experiments, even if they are unrealistic, can be instrumental in demonstrating and understanding a phenomenon. My main issue is that this paper doesn't have enough realistic settings, and I do not think that the current version of the paper crosses the line for a sufficient number of realistic benchmarks.

---

> > > > > > ### Author Response · Authors · 2024-12-02
> > > > > >
> > > > > > Thank you for your response. We would like to let you know that we have finished running experiments with Vision Transformers on the Incremental CIFAR-100 setting described in [1]. The main results are summarized in Global Response #2, and they are all generally in line with our previous conclusions. We thank the reviewer for pushing us to do these experiments as they did make the evidence for our conclusions stronger. We now have experiments that provide coverage on two realistic data shifts (Noisy and Incremental CIFAR-100) and two realistic architectures (ResNets and Vision Transformers). If we have addressed your concerns about our work, we ask that you please support our work further by raising your score.
> > > > > >
> > > > > > [1] Dohare, Shibhansh, et al. "Loss of plasticity in deep continual learning." Nature 632.8026 (2024): 768-774.

---

> > > > > > > ### Author Response · Authors · 2024-12-02
> > > > > > >
> > > > > > > Dear Reviewer,
> > > > > > >
> > > > > > > Since this is the last day where reviewers are allowed to communicate with the authors, we kindly ask you to let us know whether our additional results and changes have addressed your concerns, and if so, to please adjust your score accordingly.
> > > > > > >
> > > > > > > Thank you,
> > > > > > > The Authors

---

> > > > > > > ### Comment · Reviewer_wsyP · 2024-12-03
> > > > > > >
> > > > > > > Thank you for your response. I appreciate the work you've put into running the new experiments and adding more realistic benchmarks.
> > > > > > >
> > > > > > > However, currently, I am not confident in the new results for a couple of reasons listed below.
> > > > > > >
> > > > > > > - Why is the performance of all the methods so poor on Incremental CIFAR-100? What is the performance of your model on the full dataset? From my knowledge, if we use l2 regularization, Adam, data augmentation, etc., with ViT, we can get close to 60% test accuracy on CIFAR-100 (for example, see Figure 5 in the Hare and Tortoise network paper). In a class incremental case, for a network resetting baseline, the average accuracy across all tasks could be close to 65%. While all the methods that you plot stay close to 40-45%. Can you plot a network resetting baseline for this experiment? That'll tell us if there is an issue with your experiment setup or if there is indeed a dramatic drop in performance in the increment CIFAR case.
> > > > > > > - Similarly, why does your ResNet-18 perform so poorly on noisy CIFAR-100? An 18-layer ResNet on CIFAR-100 can reach 75%+ test accuracy (for example, see Extended Data Fig. 1 in Dohare et al. 2024). Your results show that all methods are between 25 and 40%. Can you plot a network resetting baseline for this experiment? Note that data augmentation can generally only explain about a 5-7% drop in test accuracy. Again, either there is an issue with your experiment setup, or there is indeed a dramatic drop in test accuracy in the noisy CIFAR case.
> > > > > > >
> > > > > > > As an additional point, I still don't understand why the permuted CIFAR-100 with the ResNet experiment is still in the paper. The ResNet performs extremely poorly in that case, as test accuracy is only about 20-30%. It's probably no better than a similary sized full connected feed forward network.

---

> > > > > > > > ### Author Response · Authors · 2024-12-04
> > > > > > > >
> > > > > > > > Hello, thank you for your response and your continued interaction. Unfortunately, we did not have time to implement the necessary changes required to show equivalence of our setting and run the requested experiments before the deadline  We address the differences between our setup and the setups in other works below:
> > > > > > > >
> > > > > > > > **[ViT performance lower on CIFAR-100]**
> > > > > > > > The main cause of this is that we are using a smaller version of the ViT used in the Hare and Tortoise paper. The one in that paper had 12 layers, whereas due to time/resource constraints, we only used 4 layers for these experiments. They also use data augmentation and a learning rate schedule with warmup. The combination of these improves performance significantly.
> > > > > > > >
> > > > > > > > **[ResNet performance on CIFAR-100]**
> > > > > > > > There are a few differences between our setup and the setup presented in Dohare et al. 2024. The first as you point out is data augmentation. The second big difference is the use of a learning rate schedule. Dohare et al use a learning rate schedule for their CIFAR-100 experiments where they decay the learning rate to by a factor of .2 three times per task. The third is that although the last split is noise free, each split has only 10% of the total data, and the dataset is not additive like Incremental CIFAR-100. All 3 of these factors have a big effect on the final accuracy.
> > > > > > > >
> > > > > > > > Unfortunately, because of time constraints, we weren’t able to run the experiments that you requested (as they require not only running experiments with the larger models, but also modifying the structure of our code to enable the extra features such as data augmentation and learning rate schedules). We hope that the explanations above can help explain the discrepancy.

---

### Author Response · Authors · 2024-11-20
**Global Response**

We thank all the reviewers for their comments and questions. We are glad that the reviewers found our work timely (2oWR), novel (wsyP, 2oWR), and that our insights could help guide the community in designing better (wsyP, 2oWR, EYTk,MdJD) and more efficient (wsyP, MdJD) experiments.

We respond to specific concerns below, and list the changes we made to the manuscript here:
- We did an overall polish of the writing, fixing and fixed typos.
- We moved the original Figure 2 (the plot showing the method rankings) to the appendix (now Figure 14). We replaced it with a bar plot showing the actual train and test accuracies achieved by each method on each setting, and a heatmap showing how well the rankings generated in each setting correlate with each other.
- We changed Figure 3 from a line plot to a bar plot and added Protocol 3 to the evaluation.
- We improved the general readability of Figure 4a by making the points smaller and differentiating the trendlines.
- We moved Figures 5d-f in the original manuscript to the Appendix (Figure 9).
- We updated Sections 4.1 (Comparing Protocols), 4.2 (The Performance of Current Methods), and 4.3 (Correlation between Training and Testing Plasticity) with more nuanced claims and discussion.
- We added a section to the Appendix describing our Statistical Bootstrapping procedure.
- In Appendix D.1, we added Figures 10 and 11 showing the effect of the number of seeds on the hyperparameter search for each method.
- In Appendix D.5, we added Figure 15 showing the relationship between train loss plasticity and test loss plasticity. We also added Figure 16 and Table 3, showing the relationship between train accuracy plasticity and test accuracy plasticity for each method.

We also would like to inform the reviewers that we have two more sets of experiments currently running, which we will update the paper with when they are finished:
- A set of experiments with CReLU where we reduce the architecture size to control for the number of parameters.
- A set of experiments structured similarly to the ResNet experiments with a Vision Transformer network.

---

### Author Response · Authors · 2024-11-25

Dear Reviewers,

We have submitted detailed responses to all the reviewers' concerns, along with a general response summarizing the changes made. With only one day remaining in the discussion period, we kindly ask for confirmation on whether our replies have addressed the reviewers' concerns, and, if so, to consider adjusting your score accordingly.

Your feedback is crucial to improving the quality of the paper, and we would greatly appreciate your engagement before the deadline.

Thank you for your time and consideration.

Best regards,
The Authors

---

### Author Response · Authors · 2024-12-02
**Global Response #2**

We thank all the reviewers for the discussion and comments on our work so far. We think that our paper has significantly improved through this process. We would like to inform you that we have finished running experiments with Vision Transformers on Incremental CIFAR-100, in a setting similar to Dohare et. al, 2024 [1] (the main difference being that we do not perform early stopping on the tasks). In this setting, the 100 classes of CIFAR-100 are divided into 20 splits, and on each task, one of the splits (i.e. 5 classes) is added to the dataset that the model is trained and evaluated on. We increase the number of training steps on each task proportionally to the amount of data in the task, similar to [1], and train 10 seeds, similar to our ResNet-18 experiments. We use a smaller version of the DeiT architecture [2]. We summarize a few key results below and share anonymized links of new figures:

**Comparison of Protocols**
We extend the analysis in Section 4.1 (Figure 3) where we evaluate how well the rankings generated by the different protocols correlate with rankings on task sequences not seen in training. As seen [here](https://ibb.co/ft4zHjr), in the Incremental CIFAR-100 setting with ViTs, Protocol 2 and Protocol 1 are approximately even (within the confidence intervals) in their ability to transfer, and Protocol 3 lags significantly behind both. This is consistent with our previous results in that even in this new setting, there is no advantage to be had by using Protocols 1 and 3.

**Performance of Current Methods**
We also extend Figures 2a and 2b with results from Incremental CIFAR-100 experiments. We see when looking at the [performances of individual methods](https://ibb.co/c6Zkm8f) that again, several of our findings from before also hold in this setting. We report the average of the metrics across all tasks. A well tuned Online baseline (with L2 regularization added), can be fairly strong. It essentially matches the best performing methods on training plasticity and outperforms CReLU and Continual Backprop on testing plasticity. Hare and Tortoise continues to be a very good method for testing plasticity, although CReLU seems to underperform on testing plasticity compared to its performance on the other settings/architectures.

We also look at [an extended version of the heatmap in Figure 2b](https://ibb.co/BThrJF5), showing the rank correlation of the method rankings on the different settings and metrics. We find again that for most settings, there isn’t a strong correlation between the rankings. Interestingly, there is a negative correlation between the method ranks for training plasticity on Incremental CIFAR-100 and testing plasticity on Incremental CIFAR-100, providing further evidence that with many of the benchmarks used to study plasticity, we should focus more on testing plasticity rather than training plasticity.

**Training vs Testing Plasticity**
Finally, we look directly at the training plasticity and testing plasticity of the different ViT configurations sampled on Incremental CIFAR-100. We look at both the [aggregated](https://ibb.co/JmgzKhf) and [non-aggregated](https://ibb.co/fMNsdHZ) correlations between training and testing plasticity. A similar trend emerges as with the other settings where there is a clear positive trend when looking at all configurations, but when focusing on just the well performing configurations, the positive correlation disappears or even turns into a negative correlation.

Note, we also have similar versions of the other figures in our paper (looking at the impact of the number of seeds, tasks, and configurations) with the ViT Incremental CIFAR-100 results included which do not change the messages of our paper, but for brevity’s sake, we leave them out of this response. They will be included in the final version of the paper. If you have any more concerns, please let us know.

[1] Dohare, Shibhansh, et al. "Loss of plasticity in deep continual learning." Nature 632.8026 (2024): 768-774.
[2] Touvron, Hugo, et al. "Training data-efficient image transformers & distillation through attention." International conference on machine learning. PMLR, 2021.

---

### Meta-Review · Area_Chair_itx9 · 2024-12-22

**Metareview:**

In this paper, the authors address experimental design for methods addressing the training of deep networks in settings that involve optimizing the model on changing data distributions, i.e. that involve non-stationarity or require "plasticity" such as continual learning or reinforcement learning.  The authors argue for more rigorous and careful evaluation to compare methods and prior work and thus enable progress.  The authors perform a comprehensive experimental study, examine existing practices and then provide recommendations.

The reviews were borderline but not high variance with two leaning accept and two leaning reject (5, 5, 6, 6).  The reviewers found the experiments extensive, the topic important and timely and the insights useful.  As weaknesses the reviewers noted that they found the experiments somewhat "unrealistic" (e.g. permuting pixels instead of down-sampling), some of the architectures inappropriate for the given task, some lack of novelty in the recommendations, some lack of clarity / inconclusiveness of results and then issues with clarity / typos in the writing.

There was significant discussion between the reviewers and the authors, and afterwards between the reviewers and the AC.  Multiple reviewers agreed to revise their scores upwards after reading the author response, which seemed compelling.  The main point of discussion between the reviewers / AC centered around the concerns regarding how realistic the distribution shift settings were.  Note, one reviewer who was leaning 6 stated that they were now leaning reject (but presumably could no longer update their score).  Another who had put 5 stated they were now leaning more towards 3 (pointing out that they realized that the authors used a much smaller network than the standard ViT in their experiments).  One reviewer found that the experiments were appropriate, while the other three found them too unrealistic, one stating "the experiments remain insufficient for a paper claiming "experimental design in non-stationary optimization" in its title."

The discussion with reviewers seemed to establish a consensus that the paper fell below the bar for acceptance.  Therefore the recommendation is to reject the paper.  However, the reviewers all found the work timely and important, and thus the authors are encouraged to strengthen the paper for a future submission.

**Additional Comments On Reviewer Discussion:**

As written above, there was discussion between the authors and reviewers.  In summary, the authors addressed concerns regarding how unrealistic the experiments were, justified model choices, responded to concerns of lack of novelty, improved the clarity and addressed typos and added new experiments using ViTs on Incremental CIFAR-100.  The reviewers increased their scores after reading these (one from 3->5 and others by one point).

The AC started a discussion regarding concerns on how realistic the experiments were.  This discussion really moved the consensus of the reviewers toward reject.  One reviewer who was leaning 6 stated that they were now leaning reject (but presumably could no longer update their score).  Another who had put 5 stated they were now leaning more towards 3 (pointing out that they realized that the authors used a much smaller network than the standard ViT in their experiments).  One reviewer found that the experiments were appropriate, while the other three found them too unrealistic, one stating "the experiments remain insufficient for a paper claiming "experimental design in non-stationary optimization" in its title

---

### Decision · Program_Chairs · 2025-01-22

Reject